# Microbial short-chain fatty acids modulate CD8+ T cell responses and improve adoptive immunotherapy for cancer

Maik Luu[1,2], Zeno Riester [2], Adrian Baldrich[2], Nicole Reichardt[3], Samantha Yuille[3], Alessandro Busetti[3], Matthias Klein[4], Anne Wempe[1], Hanna Leister[1], Hartmann Raifer[5], Felix Picard[1], Khalid Muhammad [6], Kim Ohl [7], Rossana Romero[1], Florence Fischer[1], Christian A. Bauer[8], Magdalena Huber [1], Thomas M. Gress[8], Matthias Lauth[9], Sophia Danhof [2], Tobias Bopp [4], Thomas Nerreter [2], Imke E. Mulder[3], Ulrich Steinhoff[1], Michael Hudecek [2,10✉] & Alexander Visekruna [1,10✉]

Emerging data demonstrate that the activity of immune cells can be modulated by microbial molecules. Here, we show that the short-chain fatty acids (SCFAs) pentanoate and butyrate enhance the anti-tumor activity of cytotoxic T lymphocytes (CTLs) and chimeric antigen receptor (CAR) T cells through metabolic and epigenetic reprograming. We show that in vitro treatment of CTLs and CAR T cells with pentanoate and butyrate increases the function of mTOR as a central cellular metabolic sensor, and inhibits class I histone deacetylase activity. This reprogramming results in elevated production of effector molecules such as CD25, IFN-γ and TNF-α, and significantly enhances the anti-tumor activity of antigen-specific CTLs and ROR1-targeting CAR T cells in syngeneic murine melanoma and pancreatic cancer models. Our data shed light onto microbial molecules that may be used for enhancing cellular anti-tumor immunity. Collectively, we identify pentanoate and butyrate as two SCFAs with therapeutic utility in the context of cellular cancer immunotherapy.

[1] Institute for Medical Microbiology and Hygiene, Philipps-University Marburg, Marburg, Germany. [2] Medizinische Klinik und Poliklinik II, Universitätsklinikum Würzburg, Würzburg, Germany. [3] 4DPharma Research Ltd., Aberdeen, United Kingdom. [4] Institute for Immunology, University Medical Center, Johannes Gutenberg University Mainz, Mainz, Germany. [5] Flow Cytometry Core Facility, Philipps University Marburg, Marburg, Germany. [6] Department of Biology, United Arab Emirates University, Al Ain, United Arab Emirates. [7] Department of Pediatrics, RWTH Aachen University, Aachen, Germany. [8] Department of Gastroenterology, Endocrinology, Metabolism and Infectiology, University Hospital Marburg, UKGM, Philipps University Marburg, Marburg, Germany. [9] Institute of Molecular Biology and Tumor Research, Center for Tumor- and Immunobiology, Philipps-University Marburg, Marburg, Germany. [10] These authors jointly supervised this work: Michael Hudecek, Alexander Visekruna. ✉email: Hudecek_M@ukw.de; alexander.visekruna@staff.uni-marburg.de

The intestinal microbiota has been shown to directly impact on the efficacy of specific cancer immune therapies[1–3]. Particularly, immune checkpoint inhibitory (ICI) therapy and adoptive cell therapy using tumor-specific CD8+ cytotoxic T lymphocytes (CTLs) can be influenced by the composition of the intestinal microbiota[4,5]. Recently, several studies have demonstrated that members of the gut microbiota are able to enhance the anti-tumor efficacy of PD-1 and CTLA4 blockade therapy[6,7]. *Akkermansia muciniphila* and some *Bifidobacterium* strains have been shown to modulate anti-tumor immune responses and improve ICI treatment[1,3]. Furthermore, a defined commensal consortium consisting of 11 human bacterial strains elicited strong CD8+ T cell-mediated anti-tumor immunity in an experimental subcutaneous tumor model[7]. This study has demonstrated that a mixture of human low-abundant commensals was able to substantially enhance the efficacy of ICI therapy in mice[7,8].

Relatively little is known about the therapeutic potential of soluble microbial molecules and metabolites to modulate the outcome of cellular cancer immunotherapy. Specific commensal bacteria have been shown to synthesize the metabolite inosine that is able to enhance the efficacy of ICI therapy[9]. Previous studies have identified acetate, propionate, and butyrate as major microbial metabolites, belonging to the class of short-chain fatty acids (SCFAs). These were shown to promote the expansion of Tregs, but they also seem to improve the function of effector T cells[10–15]. Butyrate is associated with protection from autoimmune processes and graft-versus-host disease[16,17]. Furthermore, two reports have highlighted the role of butyrate in promoting the memory potential and antiviral cytotoxic effector functions of CD8+ T cells[18,19]. We have recently demonstrated that the SCFA pentanoate (valerate) is also a bacterial metabolite present in the gut of conventional but not of germ-free (GF) mice[20]. Interestingly, dominant commensal bacteria are not able to produce pentanoate. This SCFA is rather a rare bacterial metabolite generated by low-abundant commensals such as *Megasphaera massiliensis*, which we have classified as a pentanoate-producing bacterial species[21]. It has thus far been unknown if the ex vivo culture of cytotoxic T lymphocytes (CTLs) —either derived from the endogenous repertoire or through genetic engineering with a T cell receptor (TCR) or chimeric antigen receptor (CAR)—with bacterial metabolites is capable of augmenting their anti-tumor reactivity.

In this study, the influence of bacterial SCFAs on adoptive T cell therapy using antigen-specific CD8+ CTLs and CAR T cells has been investigated. We demonstrate that the treatment with pentanoate and butyrate induces strong production of effector molecules in CTLs and CAR T cells, resulting in increased anti-tumor reactivity and improved therapeutic outcome. These data show that specific gut microbiota-derived metabolites such as butyrate and pentanoate have the potential to optimize adoptive T cell therapy for cancer in humans.

## Results

### *Megasphaera massiliensis*-derived pentanoate enhances the production of effector cytokines in CD8+ T cells. 

Commensal bacteria harboring a broad spectrum of enzymes have the ability to produce a large variety of small molecules that may be exploited for therapeutic interventions. Recently, we found that a human gut-isolated bacterial strain *Megasphaera massiliensis* was able to produce high levels of the SCFA pentanoate[21]. Previously, we showed that pentanoate substantially influenced the function of CD4+ T lymphocytes by altering their epigenetic status via inhibition of histone deacetylases (HDACs)[20]. When compared to 14 abundant bacterial species, which represent proportional distribution of the most common phyla in the human intestine

(Firmicutes: *Enterococcus faecalis*, *Faecalibacterium prausnitzii*, *Anaerostipes hadrus*, *Blautia coccoides*, *Dorea longicatena*, *Faecalicatena contorta* and *Ruminococcus gnavus*; Bacteroidetes: *Bacteroides fragilis*, *Parabacteroides distasonis*, *Bacteroides vulgatus*, and *Bacteroides ovatus*; Actinobacteria: *Bifidobacterium longum* and *Bifidobacterium breve*; and Proteobacteria: *Escherichia coli*), we observed that the low-abundant human commensal *M. massiliensis* was the only bacterium synthesizing high amounts of pentanoate (Fig. 1a). Interestingly, gas chromatography-mass spectrometry (GC-MS) analysis revealed that, in addition to two different *M. massiliensis* strains (DSM 26228 and NCIMB 42787), *Megasphaera elsdenii*, which is the closest phylogenetic neighbor of *M. massiliensis*[22], also produced pentanoate, although not at such high levels as *M. massiliensis*. Of note, two *M. massiliensis* strains were the only strong producers of the SCFAs pentanoate and butyrate. While most commensals generated high amounts of acetate and formate, *Faecalibacterium prausnitzii* and *Anaerostipes hadrus* synthesized high levels of butyrate (Fig. 1a).

We next investigated if the pentanoate- and butyrate-containing supernatant from *M. massiliensis* has any impact on the function of CD8+ T cells. Indeed, the frequency of IFN-γ+TNF-α+CD8+ T cells within CTLs was markedly increased after treatment with the *M. massiliensis*-derived supernatants (Fig. 1b). For comparison, we included supernatants from several highly abundant gut commensal bacteria (e.g,. *E. coli*, *E. faecalis*, and *B. fragilis*) and did not observe an effect on TNF-α secretion (Supplementary Fig. 1a), consistent with their lacking ability to produce pentanoate and butyrate (Fig. 1a). Because commensal bacteria produce various soluble molecules, we cannot exclude the involvement of other microbial metabolites apart from SCFAs in observed phenomenon. To elucidate the direct SCFA-mediated effects on CD8+ T cells, we cultivated CTLs in the presence of acetate, propionate, butyrate or pentanoate. We found that among the investigated SCFAs, especially pentanoate and butyrate triggered a substantial increase in frequencies of TNF-α+ IFN-γ+ cells and secretion of TNF-α by CTLs (Fig. 1c, d). We exposed CTLs to a wide range of SCFA concentrations to select optimal concentrations promoting strong immunostimulatory effects with the least toxicity (Supplementary Fig. 2a–c). As pentanoate and butyrate promoted the most pronounced effects on CTLs, we decided to perform further functional analysis with these two SCFAs.

### Butyrate and pentanoate inhibit HDAC class I enzymes and increase mTOR activity in CD8+ T cells. 

SCFAs are known inhibitors of histone deacetylases (HDACs) enzymes, that are able to epigenetically modulate the fate of eukaryotic cells[23]. Our broad screening approach revealed that among the commensal strains tested, only a few bacterial species exhibited a strong pan-HDAC inhibitory capacity[21]. When we compared the HDAC inhibitory activity of supernatants derived from 16 human commensals, we observed that particularly the HDAC class I inhibitory effect was mediated by pentanoate- and butyrate-generating *M. massiliensis* and *M. elsdenii*, as well as by the butyrate producers *F. prausnitzii* and *A. hadrus* (Supplementary Fig. 1b), but not by other bacteria. Moreover, the activity of HDAC class I isoforms was strongly inhibited by propionate, butyrate and pentanoate. This effect was specific, since class II HDACs (HDAC4, HDAC5, HDAC6, and HDAC9) were not affected following treatment with SCFAs. As expected, the pan-HDAC inhibitor TSA inhibited both, class I and class II HDACs (Fig. 2a). Similarly, CTL-derived cell lysates exposed to propionate, butyrate, and pentanoate, but not to acetate and hexanoate, displayed a strong reduction of pan-HDAC activity (Fig. 2b). We

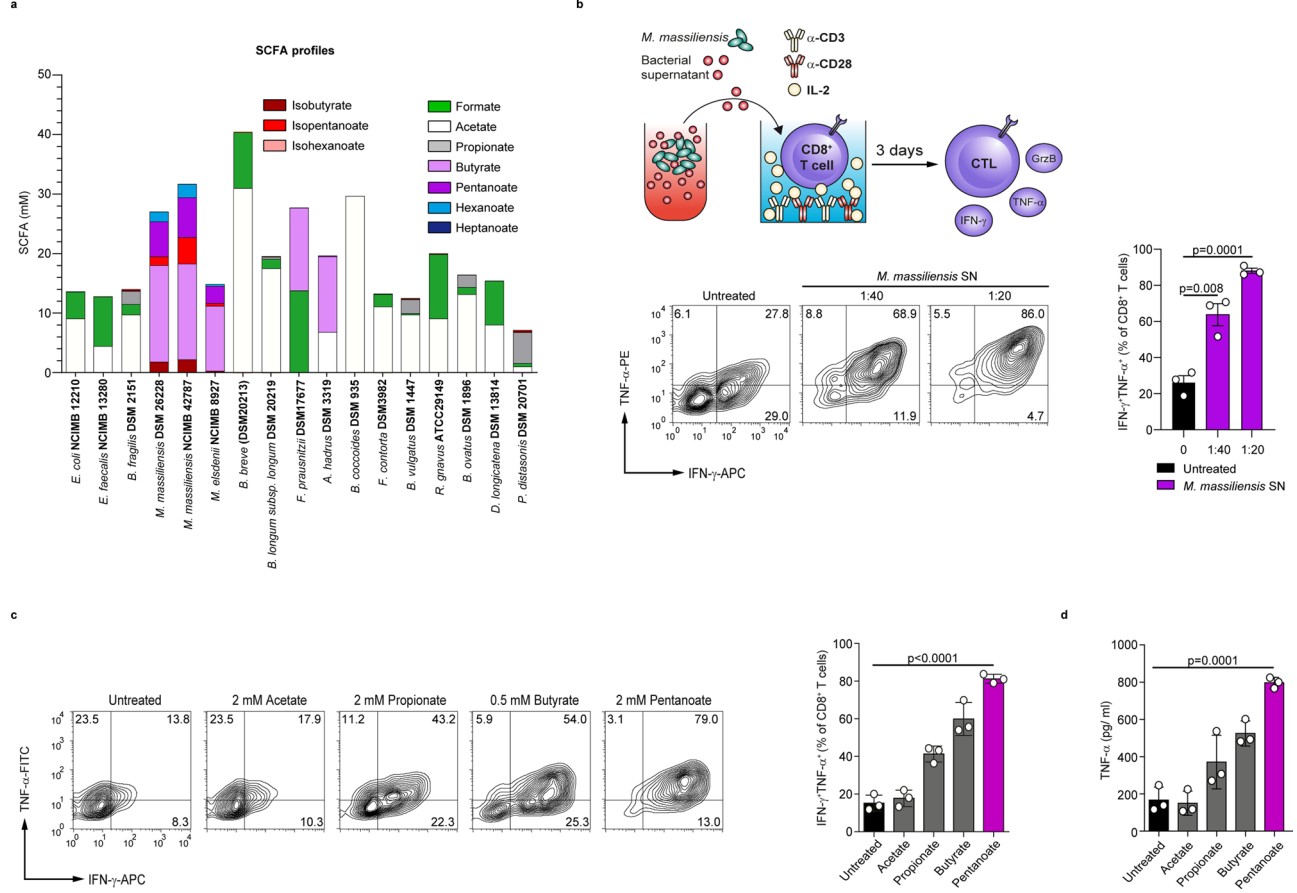

**Fig. 1 Microbiota-derived SCFAs promote the production of effector cytokines in CTLs. a** The production of SCFAs, branched-chain fatty acids (BCFAs) and medium-chain fatty acids (MCFAs) by 16 human commensals was measured by GC-MS. All bacteria were grown in vitro until the stationary growth phase before the measurement of fatty acids in supernatants ($n = 3$ independent experiments). **b** The frequency of IFN-γ- and TNF-α-expressing CD8$^+$ T cells cultured under suboptimal CTL conditions and stimulated with supernatant derived from *M. massiliensis* (1:40 or 1:20 supernatant-to-cell media ratios; $n = 3$ mice). **c** The percentage of IFN-γ$^+$TNF-α$^+$ CTLs, derived from SPF mice and treated with the indicated SCFAs for three days ($n = 3$ mice). **d** The secretion of TNF-α from CTLs treated with SCFAs was determined by ELISA ($n = 3$ mice). Statistical analysis was performed by two-tailed unpaired Student's *t*-test; mean ± s.e.m. values are presented. Source data are provided as a Source data file.

further observed that the pan-HDAC inhibitor trichostatin A (TSA) increased the percentages of IFN-γ$^+$ CD8$^+$ T cells as compared to control CTLs (Supplementary Fig. 3a). Also valproate, which is a synthetic branched SCFA derived from pentanoate with a strong HDAC inhibitory activity, potently enhanced the production of TNF-α and IFN-γ in CTLs (Supplementary Fig. 3b, c). Both pentanoate and valproate induced the expression of CTL-related transcription factors T-bet and Eomes in CD8$^+$ T cells (Supplementary Fig. 3d, e). Mocetinostat, a specific inhibitor of class I HDACs, also increased the percentage of IFN-γ$^+$TNF-α$^+$ cells and secretion of TNF-α by CTLs. In contrast, the class II HDAC inhibitor TMP-195 showed no significant effect (Fig. 2c–e). Together, these results show that the HDAC class I inhibitory activity of SCFAs pentanoate and butyrate modulates the expression of several effector molecules in CTLs.

It is known that the glycolytic metabolic pathway promotes IFN-γ expression and T cell effector function[20,24–27]. In line with these findings, inhibition of glycolysis by the glucose analog 2-deoxyglucose (2-DG) or the mTOR complex (which promotes glycolytic metabolism in effector T cells) by rapamycin led to a reduction in IFN-γ production in CTLs (Supplementary Fig. 4a). Since microbial metabolites can be utilized by T cells for their metabolic demand to enhance glycolysis and oxidative phosphorylation[28,29], we tested if pentanoate is capable of

increasing the activity of the mTOR complex, a key regulator of cell growth and immunometabolism. Indeed, pentanoate elevated the phosphorylation levels of both mTOR and its downstream target S6 ribosomal protein in CTLs. Interestingly, neither mocetinostat nor TMP-195 had a significant effect on the phosphorylation of mTOR and S6, suggesting a HDAC-independent impact of pentanoate and butyrate on metabolic status of CD8$^+$ T cells (Fig. 2f). Moreover, the extracellular acidification rate (ECAR), as an indicator of glycolytic metabolism, increased upon pentanoate treatment of CTLs (Supplementary Fig. 4b). Notably, the co-treatment of CTLs with rapamycin led to partial reduction of pentanoate-triggered IFN-γ production (Supplementary Fig. 4c). SCFAs are not only HDAC inhibitors and metabolically active substance, but also ligands for G-protein-coupled receptors GPR41 (FFAR3) and GPR43 (FFAR2). Therefore, we tested the IFN-γ expression in CTLs lacking these two SCFA-receptors (*Ffar2$^{-/-}$Ffar3$^{-/-}$* CTLs) following pentanoate treatment. We did not observe any effect of GPR41 and GPR43 on the frequency of IFN-γ$^+$CD8$^+$ T cells (Supplementary Fig. 5a, b). Finally, the pentanoate-mediated inhibition of HDAC activity in CTL-derived cell lysates was not affected in the absence of GPR41 and GPR43 (Supplementary Fig. 5c). In summary, our data show that pentanoate modulates the effector molecule expression in CTLs through HDAC-inhibition and metabolic modulation in a GPR41/GPR43-independent manner.

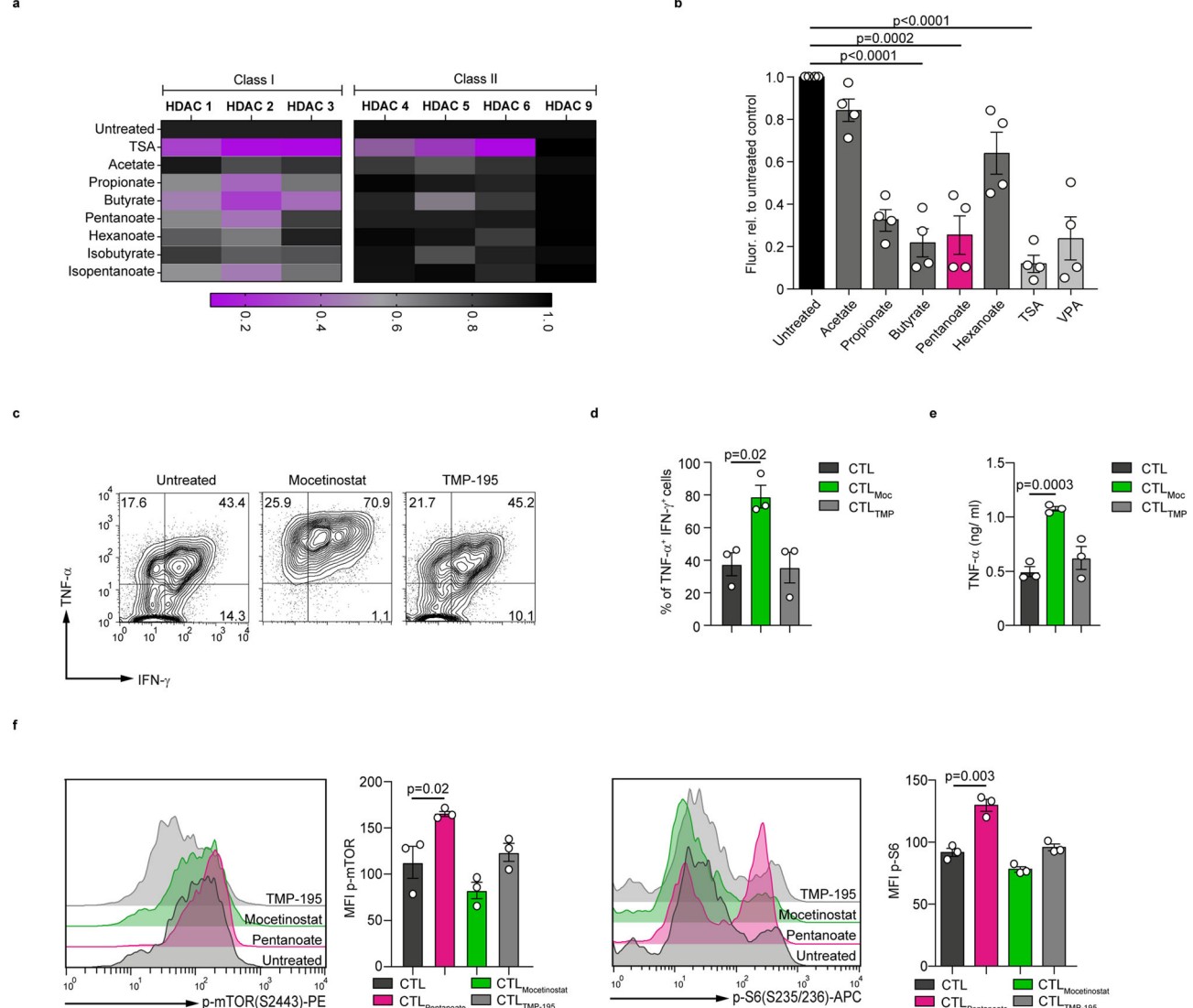

**Fig. 2 Pentanoate inhibits the class I HDAC enzymes and enhances the activity of mTOR in CD8$^+$ T cells. a** Impact of bacterial SCFAs, BCFAs, and MCFAs on the activity of recombinant class I and II HDAC enzymes. TSA was used as a control pan-HDAC inhibitor. One of three independent experiments is shown. **b** The fluorogenic HDAC assay was applied to measure the HDAC inhibitory activity of SCFAs on CTLs. The value for unstimulated CTLs was arbitrarily set to 1.0. Four independent experiments were performed (analyzed by two-tailed unpaired Student's $t$-test; data are shown as mean ± s.e.m). **c**–**e** Murine CD8$^+$ T cells were polarized under suboptimal CTL-inducing conditions for three days. Representative contour plots (**c**) and bar graphs (**d**) indicate the frequency of IFN-γ$^+$TNF-α$^+$ cells and TNF-α secretion (**e**) after treatment with 300 nM mocetinostat (class I HDAC inhibitor) or 2.5 μM TMP-195 (class II HDAC inhibitor), respectively ($n = 3$ mice, analyzed by two-tailed unpaired Student's $t$-test; data are shown as mean ± s.e.m.). **f** Murine CTLs were cultured in medium containing 1.0% serum and treated with indicated HDACi for three days. Representative histogram plots and bar graphs indicate the phoshorylated levels of mTOR and S6 ribosomal protein, respectively ($n = 3$ mice, analyzed by two-tailed unpaired Student's $t$-test; data are shown as mean ± s.e.m). Source data are provided as a Source data file.

**Pentanoate and butyrate treatment enhances the anti-tumor reactivity of antigen-specific CD8$^+$ T cells.** We next sought to assess the effect of pentanoate and butyrate treatment on the anti-tumor reactivity of antigen-specific CD8$^+$ T cells in vivo. In a first set of experiments, we injected s.c. B16OVA melanoma cells into CD45.2$^+$ mice and transferred either pentanoate- or butyrate or non-treated CD45.1$^+$ OT-I CTLs into recipient animals on day 5 after tumor injection. Overall, the anti-tumor immunity mediated by antigen-specific CTLs was significantly improved after ex vivo culture with pentanoate or butyrate as shown by decreased tumor volume and weight (Fig. 3a–c). On day 10 after adoptive transfer of CTLs, we detected a higher percentage as well as absolute number of pentanoate- and butyrate-pretreated CD8$^+$ T cells, expressing more TNF-α and IFN-γ in comparison to non-treated

OT-I CTLs in draining LNs (Fig. 3d–g). Similarly, pretreatment of CD45.1$^+$ OT-I CTLs with the supernatant of *M. massiliensis* also led to superior anti-tumor reactivity after adoptive transfer (Fig. 3b–g). Therefore, we examined if the specific HDAC class I inhibitor mocetinostat was capable of enhancing CTL-mediated anti-cancer immunity. Indeed, mocetinostat elicited a similar effect on adoptively transferred CTLs, but not to the same extent as butyrate and pentanoate, while the class II HDAC inhibitor TMP-195 did not enhance CTL-mediated anti-tumor immune responses (Fig. 3b–g). In a second set of experiments, we assessed the outcome of pentanoate pretreatment of antigen-specific CTLs in an aggressive pancreatic tumor model expressing OVA protein (OVA-expressing Panc02 cells, PancOVA). To this end, $1.5 \times 10^6$ PancOVA cells were injected s.c. into CD45.2$^+$ mice. The transfer

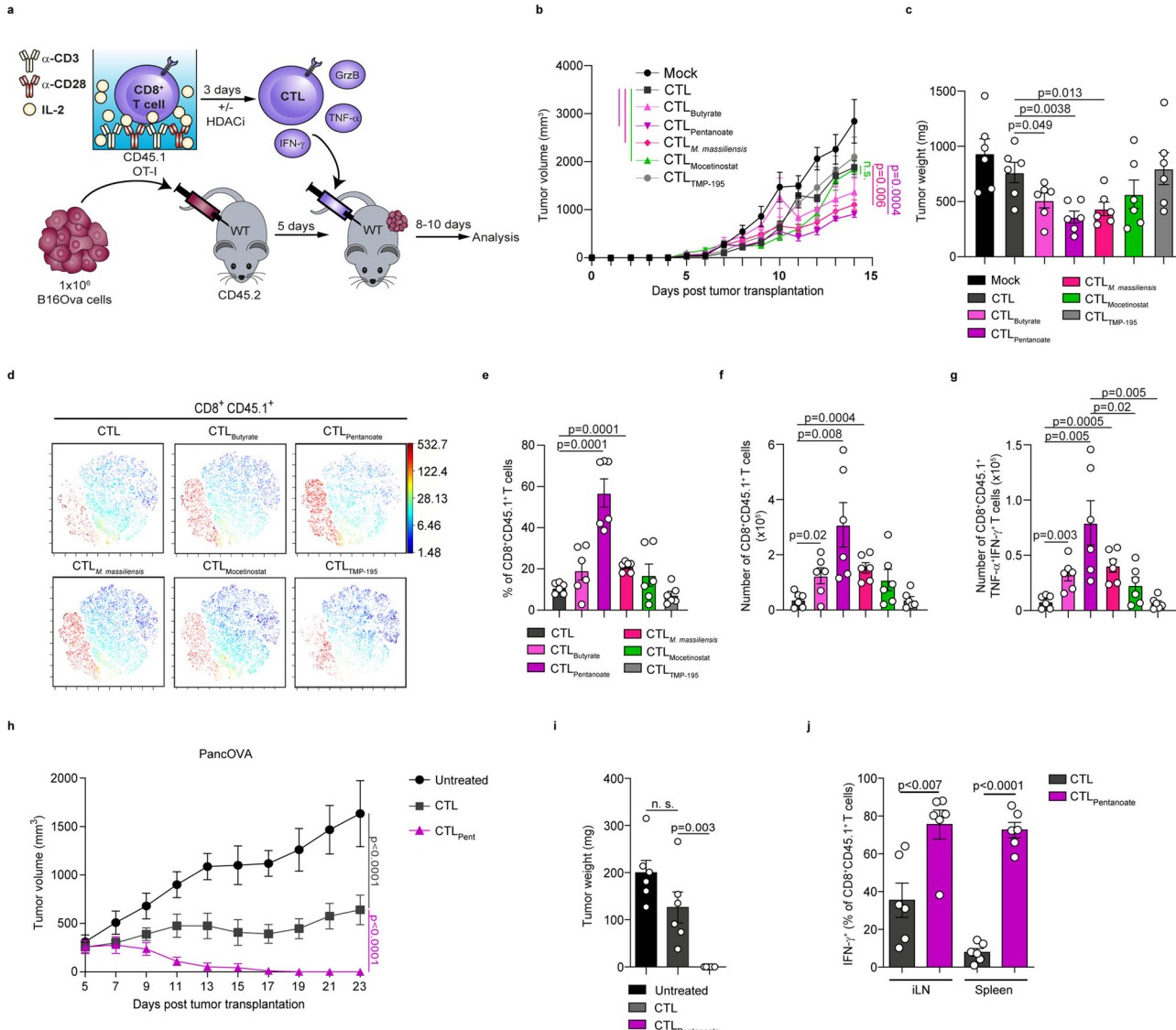

**Fig. 3 Pentanoate enhances anti-tumor activity of antigen-specific CTLs. a** Experimental design for the role of SCFAs in promoting anti-tumor immunity. **b–g** After three days of pretreatment with indicated HDAC inhibitors, CD8+CD45.1+ OVA-specific CTLs were transferred intraperitoneally (i.p.) into CD45.2+ mice bearing 5-days old B16OVA tumors. Tumor volume (**b**) and tumor mass (**c**) were analyzed on day 14 after inoculation of tumor cells (n = 6 mice/group combined from two independent experiments). The t-SNE plots in (**d**) show CD8+CD45.1+ cells among lymphocytes from the tumor-draining LNs. The percentage (**e**) and total cell number (**f**) of transferred CD8+CD45.1+ OT-I CTLs in the tumor-draining LNs at day 14 after tumor inoculation are shown. In **g** the total cell numbers of transferred antigen-specific IFN-γ+TNF-α+ CTLs were examined on day 14 after inoculation of B16OVA tumors in tumor-draining LNs (in **e–g**, n = 6 mice/group combined from two independent experiments). **h–j** OVA-specific CD45.1+ CTLs pretreated with pentanoate were adoptively transferred into CD45.2+ mice bearing 5-days old PancOVA tumors. Tumor volume (**h**) and tumor weight (**i**) were determined on day 23 post tumor inoculation. In **j** frequencies of transferred IFN-γ-producing CD8+ T cells in draining LNs and spleens were analyzed on day 23 post tumor inoculation (n = 6 mice/group combined from two independent experiments). Multiple group comparison (**b**, **h**) was performed by a linear-mixed effects model with Tukey correction. The two-tailed unpaired Student's t-test was applied to compare two groups (n.s. = not significant; results are shown as mean ± s.e.m). Source data are provided as a Source data file.

of low numbers ($7.5 \times 10^5$) of pentanoate-treated CD45.1+ OT-I CD8+ T lymphocytes, but not of untreated ones, was able to abrogate the growth of PancOVA tumor cells in recipient animals (Fig. 3h–j). While the control CTLs were hardly found on day 23 after tumor inoculation, we observed a persistence of pentanoate-treated CTLs with high IFN-γ production within draining LNs and spleen of recipient mice even 10 days after first achievement of remission (Fig. 3j and Supplementary Fig. 6a, b). These data suggest that SCFAs pentanoate and butyrate may used to augment adoptive cell therapy to target tumors. However, it should be noticed that in vivo administration of pentanoate did not enhance anti-tumor immunity. Similarly, there was no beneficial

effect of pentanoate on anti-PD-1 therapy (Supplementary Fig. 7a, b). In accordance with this observation, a recent study has revealed that the systemic administration of butyrate did not promote beneficial effects. Interestingly, butyrate even limited the effect of CTLA-4 blockade by inhibiting upregulation of CD80 and CD86 on dendritic cells[30].

**Pentanoate promotes the expression of CD25 and IL-2 in CD8+ T cells.** To investigate the capacity of SCFAs on survival and persistence of CD8+ T cells, we mixed pentanoate-treated CTLs (CD45.2+) with control CTLs (CD45.1+) at 1:1 ratio and co-

transferred them into Rag1-deficient mice. To better mimic the in vivo tumor microenvironment, we also adoptively co-transferred Tregs (CD45.2$^+$ from FIR × tiger reporter mice) that are frequently found in solid tumors. In addition, Foxp3$^-$CD4$^+$ T cells were co-transferred as the cellular source of IL-2 (Supplementary Fig. 8a, b). Surprisingly, in contrast to control CTLs, pentanoate-treated CTLs were found at a high cell number and frequency on day 15 after transfer, even without encounter of the cognate antigen (Supplementary Fig. 8c, d). The capacity of CFSE to label proliferative cells was used for in vivo monitoring of CD8$^+$ T cell proliferation in Rag1-deficient mice after pretreatment with pentanoate. We found that pentanoate-treated and CFSE-labeled CD8$^+$ T cells had a stronger proliferation as compared to control CTLs (Supplementary Fig. 8e). Furthermore, we detected higher frequencies and cell numbers of CD25$^+$ CTLs pretreated with pentanoate in Rag1-deficient mice (Supplementary Fig. 8f, g). Given the importance of the CD25 and IL-2 signaling in supporting proliferation and survival of lymphocytes[31], we next asked whether pentanoate is capable of modulating CD25 expression and IL-2 signaling in CTLs. By analyzing effects of pentanoate on CD25 upregulation and dynamics of IL-2-induced phosphorylation of STAT5, we found a higher expression of CD25 as well as a stronger STAT5 activity in response to IL-2 in pentanoate-treated CTLs in comparison to control cells (Supplementary Fig. 8h, i). Notably, the pharmacological inhibition of glycolysis with 2-DG completely abrogated pentanoate-mediated upregulation of CD25 (Supplementary Fig. 8h). Finally, pentanoate-treated CTLs robustly produced IL-2 in a prolonged manner as compared to control CTLs (Supplementary Fig. 8j), thus being able to sustain this autocrine loop for a longer period of time. Taken together, the pentanoate-induced upregulation of CD25, as well as continuous IL-2 production by CTLs, might contribute to their persistence in vivo.

**Pentanoate treatment augments the anti-tumor potency of CAR T cells in a pancreatic cancer model.** To gain further insight into potential therapeutic strategies, we examined the impact of SCFAs on genetically engineered chimeric antigen receptor (CAR) T cells. For this purpose, we used murine CD8$^+$ CAR T cells that recognize receptor tyrosine kinase-like orphan receptor 1 (ROR1) (Fig. 4a), a molecule frequently expressed in a variety of epithelial tumors and in some B cell malignancies[32]. Similar to our previous observations with CTLs, ROR1-CAR T cells that were treated with butyrate or pentanoate also enhanced the expression of CD25 upon stimulation (Fig. 4b). Moreover, ROR1-CAR T cells treated with butyrate or pentanoate enhanced their TNF-α and IFN-γ production as compared to non-treated ROR1-CAR T cells (Fig. 4c, d). We further sought to determine, whether pentanoate pretreatment of ROR1-CAR T cells had a supportive effect on their anti-tumor reactivity in a pancreatic cancer model. Therefore, we generated ROR1-expressing Panc02 pancreatic tumor cells (Panc02/ROR1) and injected them s.c. into WT mice (Fig. 4a). On day 5 after tumor cell injection, we adoptively transferred pentanoate-treated or untreated ROR1-CAR T cells and monitored T cell engraftment as well as tumor response. By day 10 following CAR T cell administration, tumor volume and weight were significantly diminished in mice that received pentanoate-treated CAR T cells as compared to animals with untreated CAR T cells (Fig. 4e, f). Furthermore, we found an elevated frequency and absolute number of IFN-γ$^+$TNF-α$^+$ pentanoate-pretreated CAR T cells in tumors, as compared to control CAR T cells (Fig. 4g).

Previously, we could demonstrate that human CD8$^+$ T lymphocytes equipped with a ROR1-CAR exert specific anti-tumor reactivity in vitro and xenograft models in

immunodeficient mice[33,34]. Hence, we investigated the impact of SCFAs on human CD8$^+$ T lymphocytes and CAR T cells. Our data show that both pentanoate and butyrate as well as the class I HDAC inhibitor mocetinostat enhanced expression of TNF-α and IFN-γ in human CD8$^+$ T cells in non-toxic concentration range (Fig. 5a and Supplementary Fig. 9a). Moreover, both SCFAs increased the phosphorylation of mTOR and S6 ribosomal protein in CTLs. In contrast to SCFAs, mocetinostat and TMP-195 were not capable of significantly activating mTOR pathway in human CTLs (Supplementary Fig. 9b, c). Based on findings collected from murine CAR T cells, in a complementary approach, we pretreated human ROR1-CAR T cells with pentanoate for 4 days and subsequently investigated the cytokine production, proliferation, expression of activation markers and the cytotoxic capacity of untreated and pentanoate-treated CAR T cells as indicated in the Fig. 5b. In accordance with results generated with murine CTLs and CAR T cells, human CAR T cells also upregulated the expression of CD25 and secretion of IL-2 after pentanoate pretreatment as compared to untreated cells (Fig. 5c, d). When we co-cultured ROR1-specific CAR T cells with ROR1-expressing K562 human cancer cells, we found that pentanoate-pretreated cells elevated the production of CTL-related cytokines IFN-γ and TNF-α (Fig. 5e). Moreover, CAR T cells pretreated with pentanoate proliferated stronger than control CAR T cells and exerted a superior cytolytic activity after encounter of ROR1 (Fig. 5f, g). Together, these results show that pentanoate and butyrate treatment of murine and human ROR1-specific CAR T cells augments their anti-tumor function in vitro and in vivo. Collectively, the data demonstrate the potential to use pentanoate and butyrate treatment during CAR T cell manufacturing to exploit the beneficial effects of these bacterial metabolites on CTL function in order to increase the therapeutic potential and the outcome after adoptive transfer of CAR T cells.

## Discussion

Gut microbiota and microbial metabolites influence many aspects of host physiology[35]. Previously, the commensal *Bifidobacterium* was shown to enhance anti-tumor immunity in mice. The eradication of established tumors was abrogated in CD8$^+$ T cell-depleted animals, indicating that the underlying mechanism was mediated through host anti-cancer CTL responses[1]. Similarly, *Akkermansia muciniphila* increased the recruitment of CD4$^+$ T cells into tumors[3]. SCFAs such as acetate, propionate, and butyrate are water-soluble and diffusible gut-microbiota-derived metabolites reaching their peak concentrations in the caecum and decrease from the proximal to the distal colon[36]. In this study, we demonstrate that the SCFAs pentanoate and butyrate are promising microbial metabolites augmenting adoptive cell therapy for cancer. Moreover, pentanoate was able to enhance the efficacy of ROR1-specific CAR T cells. SCFAs have been shown to impact on T cell immune responses[37]. Particularly, butyrate and propionate seem to be potent regulators of the suppressive activity of Tregs in diverse experimental settings of autoimmune and inflammatory disorders[16,38]. Commensal-derived butyrate was shown to induce epigenetic reprogramming of Tregs by enhancing the acetylation of histone H3 in the promoter and conserved non-coding sequence regions of the *Foxp3* locus[12]. On the other side, some reports suggest that SCFAs are also capable of promoting the activity of CD4$^+$ effector T cells, indicating that the influence of microbial metabolites on the immune system may be more complex than previously thought[13,15]. Two recent studies have demonstrated that SCFAs were able to affect the effector function of CD8$^+$ T cells and their transition into memory cells during viral infection[18,19].

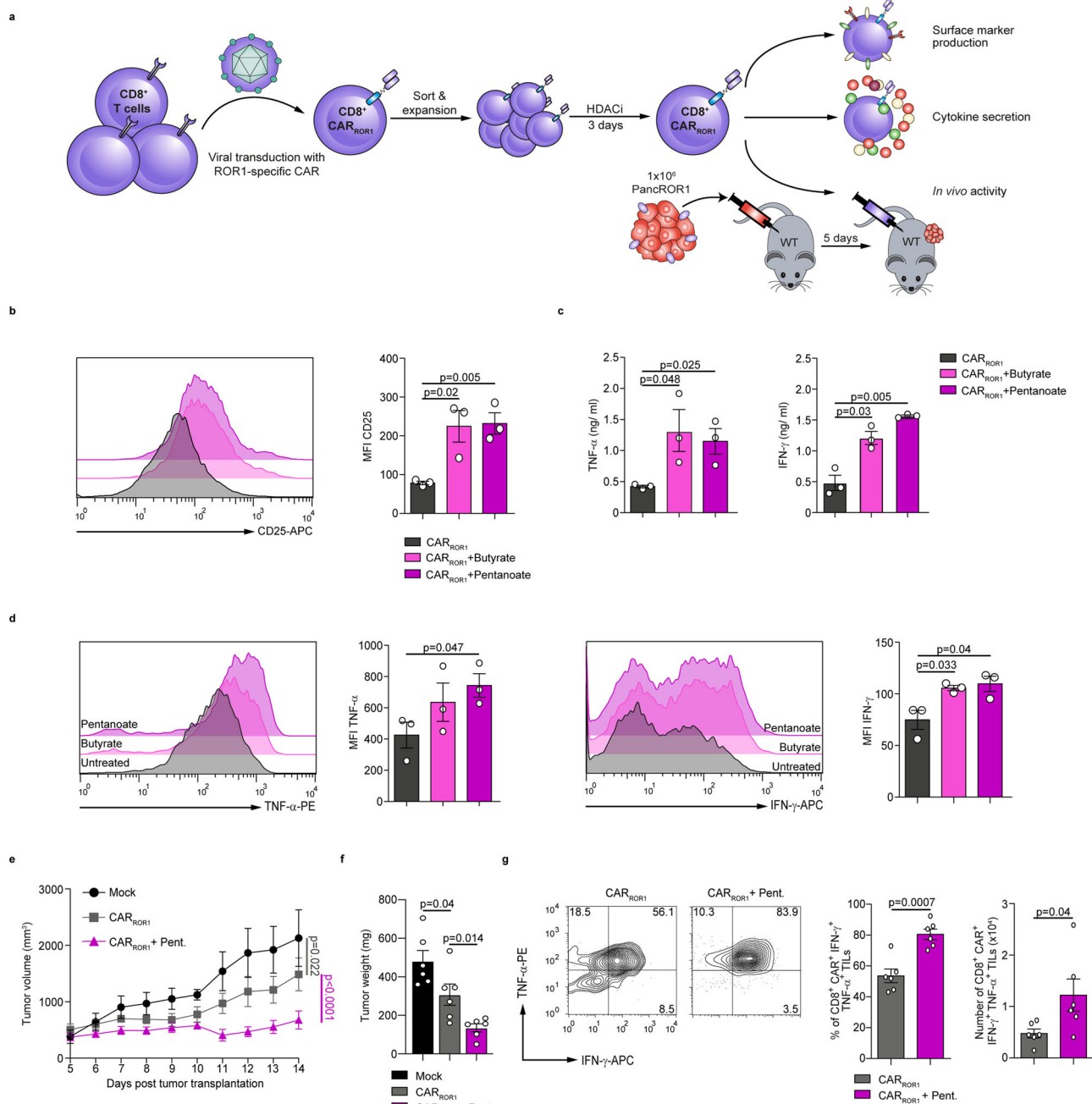

**Fig. 4 Pentanoate-treatment enhances the anti-tumor activity of murine CAR T cells. a** Experimental design for the analysis of SCFA-treated ROR1-specific murine CD8$^+$ CAR T cells (CAR$_{ROR1}$). **b** The surface expression of CD25 on CAR T cells was measured by flow cytometry analysis after three days of stimulation with butyrate or pentanoate ($n = 3$ independent experiments). **c, d** The production of TNF-α and IFN-γ from CAR T cells was measured by ELISA (**c**) and flow cytometry analysis (**d**) after three days of stimulation with butyrate or pentanoate. Three similar experiments were performed. **e–g** Following the pretreatment with pentanoate for three days, ROR1-specific CAR T cells were transferred i.p. into mice bearing 5-days old PancROR1 tumors. Tumor volume (**e**) and tumor mass (**f**) were analyzed on day 14 post tumor inoculation ($n = 6$ mice/group combined from two independent experiments). The percentage and total cell number of transferred TNF-α$^+$ and IFN-γ$^+$ CAR T cells in the tumor tissue at day 14 after tumor inoculation are shown in (**g**). For all experiments, 0.75 mM butyrate and 2.5 mM pentanoate were used, respectively. Multiple group comparison in (**e**) was performed by a linear-mixed effects model with Tukey correction. In **b–d** and **f, g**, the two-tailed unpaired Student's t-test was applied. Source data are provided as a Source data file.

Another potentially therapeutically useful SCFA, pentanoate, has not been investigated yet in the context of cancer immunotherapy. We recently found that pentanoate was able to impact on the function of CD4$^+$ effector T cells by modulating both cellular metabolism through the activation of mTOR pathway and epigenetic status of cells by inhibiting HDAC enzymes[20]. Our current findings have revealed that pentanoate and butyrate promoted their effects on CD8$^+$ T cells primarily by inhibiting class I HDAC enzymes and by inducing metabolic alterations, which resulted in enhancement of the expression of CTL-associated genes. Our recent report demonstrated that the T cell-specific deletion of class I HDACs HDAC1 and HDAC2 (a deletion of two *Hdac1* alleles and one *Hdac2* allele) triggered the expression of genes associated with cytolytic activity even in CD4$^+$ T cells[39]. Furthermore, pentanoate

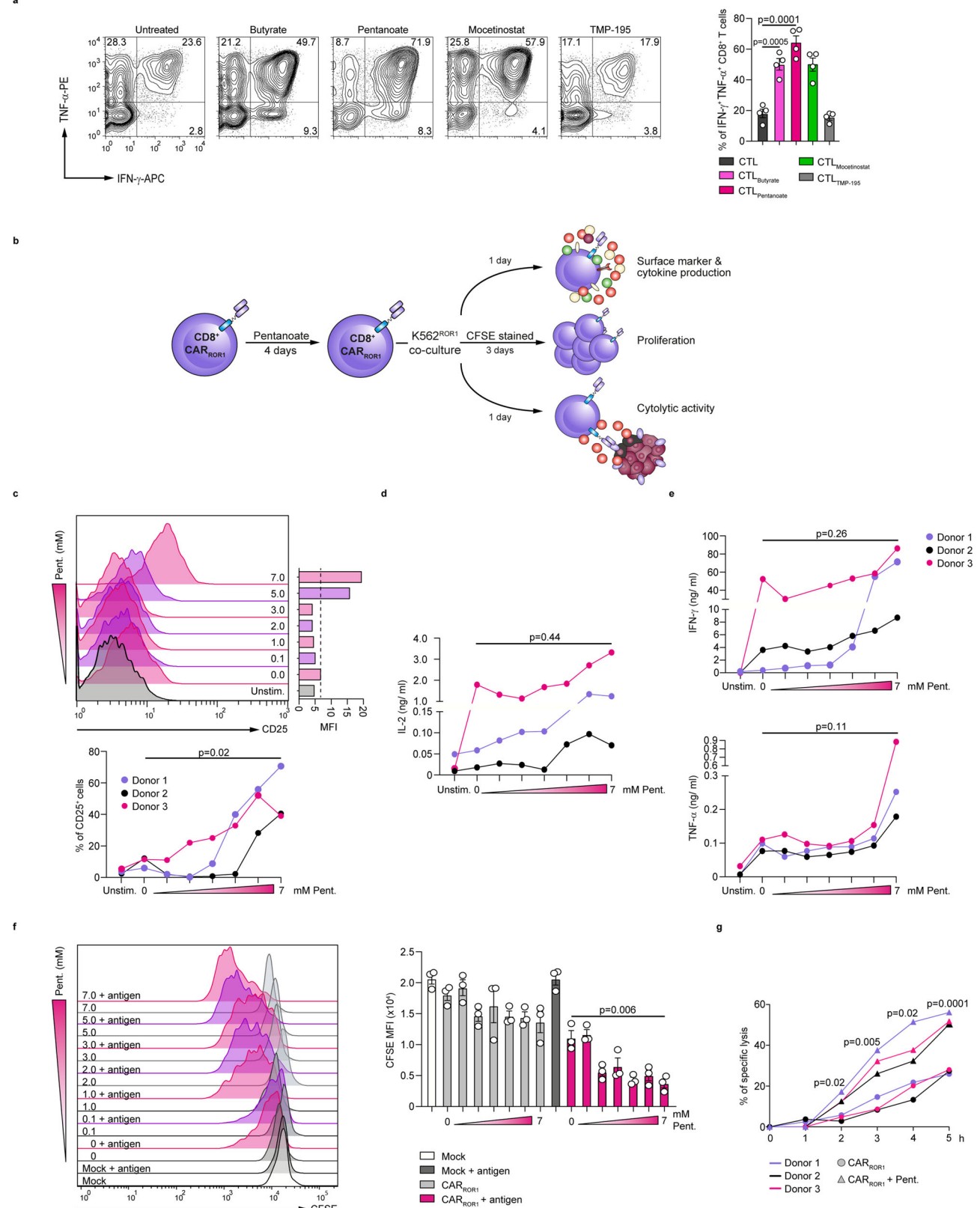

was also able to induce in vitro CD4+ CTL differentiation in human CD4+ T cells with similar phenotype[39], suggesting a predominant role for HDAC inhibitory activity over metabolic effects for SCFA-treated T cells. Interestingly, although the specific class I and class II HDAC inhibitors mocetinostat and TMP-195 were not capable of increasing the phosphorylation of mTOR and its target

protein S6, the pan-HDAC inhibitor TSA was shown to increase mTOR activity in T cells[13], suggesting a certain level of interdependence between HDAC inhibition and metabolic effects. In the future, the complex cross-talk between SCFA-mediated epigenetic changes and their multiple effects on the cellular metabolism of CTLs should be investigated in more detail.

**Fig. 5 Pentanoate enhances the functional status of human CAR T cells. a** CD8$^+$ T cells were isolated from peripheral blood of healthy donors were differentiated into CTLs in presence or absence of butyrate (1 mM), pentanoate (4 mM), mocetinostat (HDAC class I inhibitor, 300 nM) or TMP-195 (HDAC class II inhibitor, 2.5 µM). Representative contour plots (left) and graphs (right) show the frequency of IFN-γ$^+$ TNF-α$^+$ CD8$^+$ T cells. (n = 4 combined from four independent experiments). **b** Experimental setup for the functional analysis of pentanoate-treated ROR1-specific human CD8$^+$ CAR T cells (CAR$_{ROR1}$ T cells). **c** The surface expression of CD25 was measured following pentanoate treatment by flow cytometry. **d** The secretion of IL-2 was detected in supernatants of pentanoate-treated CAR$_{ROR1}$ T cells by ELISA. **e** The secretion of IFN-γ and TNF-α was analyzed in supernatants of pentanoate-treated CAR$_{ROR1}$ T cells by ELISA. **f, g** Proliferation of CAR$_{ROR1}$ T cells was determined by CFSE labeling (**f**). CAR$_{ROR1}$ T cells pretreated with pentanoate were stained with CFSE and subsequently co-cultured with K562$^{ROR1}$ cells in the absence of pentanoate. CD8$^+$ T cells without the CAR construct were used as mock control cells. The cytolytic activity of CAR$_{ROR1}$ T cells was examined by analysis of specific lysis following encounter with luciferase-expressing K562$^{ROR1}$ cells (**g**). The percentage of lysed target cells was determined in 1 h intervals (effector-to-target cell (E:T) ratio = 2.5:1). Data points shown in the graphs represent CAR$_{ROR1}$ T cells derived from three different donors (**c–g**). Following pentanoate pretreatment, the stimulation was mediated by co-culture of CD8$^+$ CAR$_{ROR1}$ T cells with ROR1-expressing K562 (K562$^{ROR1}$) cells in the absence of pentanoate. The groups in **a** and **c–g** were compared by two-tailed unpaired Student's *t*-test, data shown as mean ± s.e.m. Source data are provided as a Source data file.

SCFAs can regulate the activity of immune cells not only by triggering metabolic and epigenetic changes, but also by binding to cognate receptors on the surface of cells. SCFAs serve as ligands for GPR41 and GPR43 on intestinal Tregs and epithelial cells, leading to activation mitogen-activated protein kinase signaling and secretion of chemokines and cytokines[11,40]. As most SCFAs have agonist activity for GPR41 and GPR43[41], we treated CTLs lacking these both SCFA-receptors with pentanoate. We did not observe any detectable influence of GPR41 and GPR43 on SCFA-mediated regulation of CTL activity. One plausible explanation might be a low expression of both molecules on T cells isolated from spleen and LNs. Only CD4$^+$ T cells derived from intestinal lamina propria, but not that from other tissues, have been demonstrate to express high levels of these SCFA-receptors[13]. We could also observe that penatoate enhanced the expression of CD25 and IL-2 in CTLs. During activation of T cells, IL-2 is consumed in an autocrine manner, which might have impact on the persistence and activity of adoptively transferred CTLs in the microenvironment of solid tumors. Interestingly, the pentanoate-mediated upregulation of CD25 in CTLs was completely abrogated upon co-treatment of cells with 2-DG, indicating that pentanoate-mediated enhancement in glycolytic activity might be responsible for observed effects. Recently, CD25 was shown to be the upstream molecule inducing mTOR signaling in T cells[42], suggesting a possible positive feedback triggered by pentanoate treatment.

Genetic modification with a CAR endows T cells with a new antigen-specificity target and eliminate tumor cells. In contrast to CD19, the first clinically approved molecule for CAR T cells, which is present on the cell surface of CD19-positive lymphoma and leukemia, the orphan receptor ROR1 is expressed on many epithelial tumors and is also a target for CAR T cells[32,43]. The potential effects of metabolites produced by the gut microbiota on the efficacy of human CAR T cells has thus far been unknown. In this study, we demonstrated that the SCFAs pentanoate and butyrate improved the efficacy of murine CAR T cells by increasing the expression of CTL-associated effector molecules in ROR1-specific CAR T cells. Most importantly, pentanoate also enhanced the efficacy of human CAR T cells suggesting that microbial metabolites could be therapeutically employed.

Our study illustrates one potential embodiment to exploit the beneficial effect of pentanoate and butyrate on CTL function, i.e. SCFA-treatment during CTL manufacturing, which can readily be implemented in clinical-grade GMP manufacturing processes[44]. Another potential embodiment is the administration of pentanoate, butyrate or other SCFAs, or the transfer of a bacterial consortium that produces these SCFAs to patients that have received adoptive cell therapy. However, the clinical implementation of this embodiment will require careful additional investigations to determine the optimal route, dosing, and schedule in order to balance the stimulation of effector vs. regulatory immune cell subsets in favor of the desired therapeutic outcome.

## Methods

**Mice**. Rag1-deficient mice, *FIR × tiger* reporter animals, and WT mice on a C57BL/6 background were maintained under SPF conditions at the Biomedical Research Center, Philipps-University of Marburg. *Ffar2$^{-/-}$Ffar3$^{-/-}$*mice (on a C57BL/6 background) were kindly provided by Dr. Stefan Offermanns (Max Planck Institute for Heart and Lung Research, Bad Nauheim, Germany). Female 8–12-week old CD45.1 OT-I and CD45.2 WT mice were used for in vivo experiments. The work in animal facility such as daily animal care, breeding and offspring separation, was carried out under SPF or GF conditions. Mice were kept at 21-23 °C and 40% humidity with a 12 h light/12 h dark light cycle. The sterile conditions for GF animals were routinely tested (twice a week) by culturing feces in thioglycollate medium under aerobic and anaerobic conditions for at least two weeks. All animal work was approved by the regional animal care and use committee of the government bureau Gießen, Hesse, Germany (Study Nr. G24/2019, Regierungspräsidium Gießen, Landgraf-Philipp-Platz 1-7, 35390 Gießen, Germany). All animal experiments were performed in accordance with the German law guidelines of animal care.

**Tumor models and adoptive transfer of T cells**. For adoptive transfer experiments, mice were subcutaneously (s.c.) injected with 1 × 10$^6$ B16-OVA or 1.5 × 10$^6$ PancOVA cells. At day 5 after tumor injection, 750,000 (in PancOVA model) or 1 × 10$^6$ (in B16OVA model) of CD45.1 OT-I CTLs were transferred intraperitoneally (i.p.) into CD45.2 mice. To analyze T cells in tumors, mice were sacrificed 10–20 days after the adoptive transfer of cells. In a subset of experiments, transferred T lymphocytes were sorted from tumors for indicated analyses.

**In vivo persistence of pentanoate-treated CTLs**. CD8$^+$ T cells expressing either CD45.1 or CD45.2 derived from congenic mice were isolated as described above and activated for three days in absence or presence of 2.5 mM pentanoate. 1.5 × 10$^6$ CD45.1$^+$ untreated control CTLs were mixed with 1.5 × 10$^6$ CD45.2$^+$ pentanoate-treated CTLs, 0.5 × 10$^6$ Foxp3$^+$ Tregs (FACS-sorted from *FIR × tiger* reporter mice) as well as 2.5 × 10$^6$ Foxp3-CD4$^+$ T cells and adoptively transferred into Rag1-deficient mice. On day 15 after the transfer, LNs and spleen were analyzed for the CTL ratio, proliferation, and expression of CD25 by FACS analysis.

**In vitro T cell differentiation**. T lymphocytes were purified from LNs and spleens of mice using the kit for negative isolation (Miltenyi Biotec) with a high purity (90–95%). Purified CD8$^+$ T cells were activated with plate-bound anti-CD3 (5 µg/ml, clone 145-2C11) and soluble anti-CD28 (1 µg/ml, clone 37.51) in the presence of 50 U/ml IL-2 and anti-IFN-γ (10 µg/ml, clone XMG1.2) to obtain suboptimal CTL conditions. In specific experiments, T cells were treated with indicated concentrations of sodium pentanoate, sodium butyrate, sodium propionate, sodium acetate, sodium valproate or TSA (all substances, Sigma-Aldrich). Furthermore, some CTLs were cultivated in the presence of 300 nM mocetinostat and 2.5 µM TMP-195, respectively. The water-soluble fraction of small intestinal, caecal, and colonic luminal content was filter-sterilized (0.2 µm filter, Millipore) and diluted (1:40) in RPMI medium before treating the CD8$^+$ T cells.

**Antibodies and flow cytometry**. After three days of the cell culture, CD4$^+$ and CD8$^+$ T lymphocytes were restimulated for 4 h with PMA (50 ng/ml)/ionomycin (750 ng/ml) in the presence of 10 mg/ml Brefeldin A (all reagents, Sigma-Aldrich). Following treatment with fluorochrome-labeled antibodies, T cells were fixed with 2% formaldehyde for intracellular cytokine staining. For Annexin V staining, T cells were washed with HBSS buffer and resuspended in a HBSS solution containing FITC-labeled AnnexinV (eBioscience). The FACS measurements were performed by ARIA III and *FACSCalibur* flow cytometers (both BD) followed by analysis using FlowJ_V10 software (TreeStar). The following antibodies were used

for the FACS staining: anti-CD3 (145-2C11, 1:300), anti-CD4 (RM4-5, 1:300), anti-CD8 (53-6.7, 1:300), anti-CD19 (clone 1D3/CD19, 1:300), anti-CD45.1 (A20, 1:300), anti-CD25 (PC61.5, 1:300), anti-IFN-γ (XMG1.2, 1:300), anti-TNF-α (MP6-XT22, 1:300) and anti-granzyme B (16G6, 1:300). Transcription factors were detected using the Foxp3 staining kit (eBioscience) as well as anti-Eomes (Dan11mag, 1:200) and anti-T-bet (eBio4B10, 1:200) antibodies. For human T cells, anti-human IFN-γ (4S.B3, 1:50), anti-human CD25 (M-A251, 1:50), anti-human CD69 (FN50, 1:50), anti-human CD8 (SK1, 1:100) and anti-human/mouse granzyme B (QA16A02, 1:50) were used. For phospho-stainings, anti-phospho-STAT5 (Tyr694) (SRBCZX, 1:20), anti-phospho-mTOR (Ser2448) (MRRBY, 1:20) and anti-phospho-S6 (Ser235/236, 1:20) (cupk43k, 1:20) were used. All antibodies were purchased from eBioscience, BD Biosciences, Invitrogen or BioLegend. The gating/sorting strategies for the FACS analysis of T cells are provided in Supplementary Fig. 10. For data collection, Cell Quest Pro version 5.1 and BD FACSDiva 6.1.3 were used. Analysis of flow cytometry data was performed with FlowJoV10.

**Intracellular phospho-staining**. CD8[+] T cells were isolated as described above and activated for three days in absence or presence of 2.5 mM pentanoate. Cells were harvested, washed in PBS and rested for 4 h in RPMI at 37 °C. After 4 h resting, cells were kinetically incubated with 50 U/ml IL-2. The reaction was stopped by washing the samples with cold PBS on ice and subsequent fixation with 2% formaldehyde solution for 10 min at 37 °C. Permeabilisation was performed for 30 min by slow addition of 98% methanol (−20 °C) to a final concentration of 90% on ice. The samples were pelletized and washed in phospho-washing buffer (PBS, 2% FCS, 0.2% Tween-20). The cells were stained by incubation with the respective antibody for 45-60 min at RT. 1 ml of phospho-washing buffer was added for 10 min at RT prior to washing and further analysis.

**Measurement of ECAR**. Murine CTLs were cultured with or without pentanoate (2.5 mM) for three days. ECAR was measured with the XF96 Analyzer (Seahorse Biosciences). $2 \times 10^5$ CTLs/well were used for ECAR analysis. Basal ECAR reading was carried out with T cells grown in base DMEM without addition of glucose. ECAR was measured under basal conditions and in response to glucose (10 mM), oligomycin (2 μM, Seahorse Biosciences), and 2-deoxy-glucose (2-DG, 100 mM).

**ELISA**. TNF-α secretion from murine T cell cultures was detected by the mouse TNF-α Mouse ELISA kit (Invitrogen) according to the manufacturer's instructions. For human CD8[+] T cells, IL-2, IFN-γ, and TNF-α secretion was detected by ELISA (Biolegend). Absorbance was measured using a FLUOstar Omega ELISA plate reader (BMG Labtech).

**Generation of a ROR1-expressing pancreatic tumor cell line**. Panc02 cells were cultured in RPMI. The cells were harvested and $1 \times 10^5$ cells in 500 μl medium with 5 μg/ml polybrene seeded in 24-well plates. Lentivirus was added to a final MOI of 10. The cells were washed 48 h after transduction and expanded for 7 days. ROR1[+] cells were sorted and expanded for experiments.

**Production of retroviral supernatant**. The R11-ROR1-specific CAR containing the IgG4 hinge-CH2-CH3, CD28 transmembrane domain, and a signaling module comprising the cytoplasmatic domains of 4-1BB and CD3z was described earlier[45]. A similar ROR1 CAR except containing the murine CD28, 4-1BB, and CD3ζ was constructed. As a transduction marker, a tCD19 marker separated from the CAR by a T2A ribosomal skip element was used. The sequence was cloned into the MP71 retroviral vector to create MP71-R11-CD19t. For production of retroviral supernatant, Platinum-E cells for retroviral packaging were co-transfected with MP71-R11-CD19t and the retroviral packaging construct pCL-10A1, using the Effectene transfection reagent (QIAGEN) according to the manufacturer's instructions. Supernatants were collected 2 and 3 days after transfection.

**Murine CAR T cell generation**. CD8[+] T lymphocytes were purified from spleens of C57/BL6 mice by positive selection with CD8a (Ly-2) Microbeads (Miltenyi Biotec) with high purity (90–95 %). T cells were activated with plate-bound anti-mouse CD3ε (2 μg/ml, clone 145-2C11) and anti-mouse CD28 (2 μg/ml clone 37.51) antibodies in presence of 50 U/ml IL-2. 24 h and 48 h after activation, T cells were transduced in retroviral supernatant supplemented with 10 μg/ml polybrene by centrifugation at 2500 rpm for 90 min at 32 °C. For expansion, CAR T cells were cultured in medium supplemented with 50 U/ml IL-2, IL-7 (10 ng/ml) and IL-15 (10 ng/ml). CAR-modified T cells were enriched by immunomagnetic selection using anti-mouse CD19-PE antibody (clone 1D3/CD19) and anti-PE microbeads (Miltenyi Biotec).

**Human CAR T cell generation**. T cells for CAR modification were isolated from the peripheral blood of healthy donors. All participants provided written informed consent to participate in research protocols approved by the Institutional Review Board of the University of Würzburg (146/17-me). Peripheral blood mononuclear cells (PBMCs) were isolated by Ficoll-Paque density gradient centrifugation. CD8[+] T cells were isolated by magnetic-bead separation (CD8[+] T cells Isolation Kit, Miltenyi Biotech). $1 \times 10^6$–$2 \times 10^6$ isolated CD8[+] T cells were stimulated with

CD3/CD28 Dynabeads (Thermo Fisher Scientific) at a 1:1 bead to cell ratio in presence of 50 U/ml IL-2. Two days post-activation, nucleofection of the cells was performed by addition of 1 μg of the sleeping beauty vector pT2/HB (Addgene #26557) containing the CAR construct and 0.5 μg minicircle DNA encoding the SB100X transposase to the transfection medium. Electroporation was performed according to the manufacturer's protocol using the 4D-Nucleofector™ (Lonza, program CL-120). The electroporated cells were then transferred to a 48-well plate with 0.9 ml pre-warmed CTL medium and a half medium change with CTL supplemented with 100 U/ml recombinant human IL-2 was performed after 3 h. CD3/CD28 Dynabeads were removed magnetically five days after transfection. Transfection efficiency was assessed the day after bead removal by flow cytometry via staining of the EGFRt transfection marker with AF647-conjugated anti-EGFRt mAb (Cetuximab, Eli Lilly; conjugated in-house, 1:100). A scheme illustrating the structure of the CAR construct is provided in Supplementary Fig. 11.

**Enrichment and antigen-independent expansion of human CAR T cells**. Two days after removal of the CD3/CD28 Dynabeads, CAR[+] T cells were enriched by positive selection of the EGFRt transfection marker via immunomagnetic bead separation. EGFRt[+] T cells were frozen or used directly for antigen-independent expansion. Following enrichment, $1 \times 10^5$ CAR T cells, $5 \times 10^6$ irradiated TM-LCL and $3 \times 10^7$ irradiated PBMCs were mixed and seeded in 25 cm² cell culture flasks with OKT3 (f.c. 30 ng/ml) in a total volume of 20 ml and incubated at 37 °C. CAR T cells were expanded for 10 days applying full and half medium changes.

**Functional analysis of human CAR T cells**. After expansion, CAR T cells were treated with increasing concentrations of pentanoate. 4 days after treatment, the cells were washed and stained with 0.1 μM CFSE prior to co-culture with irradiated (80 Gy) ROR1-expressing K562 target cells at 4:1 E:T ratio for 72 h. Finally, proliferation was analyzed by flow cytometry. In order to determine cytokine secretion and surface marker expression, pretreated CAR T cells were co-cultured with target cells for 24 h. at 4:1 E:T ratio at 37 °C. Subsequently, the production of IL-2, IFN-γ, and TNF-α was analyzed in the supernatants by ELISA (Biolegend) according to the factory protocol. Surface marker expression was investigated by antibody staining followed by flow cytometry. The cytotoxic capacity of CAR T cells was determined by using a biophotonic assay based on the lysis of firefly luciferase/GFP (ffluc/GFP) transduced target cells. CAR T cells and $5 \times 10^3$ ffluc[+]/GFP[+] target cells were plated in different E:T ratios in 96-well white flat bottom plates with LCL medium containing 150 μg/ml D-luciferin substrate. The plate was incubated for 24 h at 37 °C while the luminescence signal was measured at different time points using the Infinite 200 PRO plate reader (Tecan, Männedorf, Switzerland). Lysis mediated by CAR T cells was calculated as the reduction of luminescence signal by effector cells compared to mock transfected cells: Specific Lysis [%] = Mean ((lysis by mock cells) − Single value (lysis by CAR T cells)/Mean (lysis by mock cells)) × 100.

**Bacterial culture and cell-free supernatant collection**. Pure cultures of *Escherichia coli* NCIMB 12210, *Enterococcus faecalis* NCIMB 13280, *Bacteroides fragilis* DSM 2151, *B. vulgatus* DSM 1447, *B. ovatus* DSM 1896, *Megasphaera massiliensis* DSM 26228, *M. elsdenii* NCIMB 8927, *M. massiliensis* NCIMB 42787, *Bifidobacterium breve* DSM 20213, *B. longum subsp. longum* DSM 20219, *Faecalibacterium prausnitzii* DSM17677, *Anaerostipes hadrus* DSM 3319, *Blautia coccoides* DSM 935, *Dorea longicatena* DSM 13814, *Parabacteroides distasonis* DSM 20701, *Faecalicatena contorta* DSM3982 and *Ruminococcus gnavus* ATCC29149 were grown anaerobically in YCFA + broth [Per litre: casein hydrolysate 10.0 g, yeast extract 2.5 g, sodium hydrogen carbonate 4.0 g, glucose 2.0 g, cellobiose 2.0 g, soluble starch 2.0 g, di-potassium hydrogen phosphate 0.45 g, potassium di-hydrogen phosphate 0.45 g, resazurin 0.001 g, L-cysteine HCl 1.0 g, ammonium sulphate 0.9 g, sodium chloride 0.9 g, magnesium sulphate 0.09 g, calcium chloride 0.09 g, haemin 0.01 g, SCFAs 3.1 ml (acetic acid 2.026 ml/L, propionic acid 0.715 ml/L, n-valeric acid 0.119 ml/L, iso-valeric acid 0.119 ml/L, Iso-butyric acid 0.119 ml/L), vitamin mix 1: 1 ml (biotin 1 mg/100 ml, cyanocobalamine 1 mg/100 ml, p-aminobenzoic acid 3 mg/100 ml, pyridoxine 15 mg/100 ml), vitamin mix 2:1 ml (thiamine 5 mg/100 ml, riboflavin 5 mg/100 ml), vitamin mix 3:1 ml (folic acid 5 mg/100 ml)] until they reached their stationary growth phase. Cultures were centrifuged at $5000 \times g$ for 10 min and the cell-free supernatant (CFS) was filtered using a 0.45 μm followed by a 0.2 μm filter (Millipore). 1 mL aliquots of the CFS were stored at −80 °C until further use. In some experiments, filter-sterilized supernatant of *Megasphaera massiliensis* DSM 26228 were used at 1:40 or 1:20 supernatant to cell media ratio to treat CTLs.

**Measurement of fatty acids in bacterial supernatants**. The composition of SCFAs and MCFAs in bacterial supernatants was determined by MS Omics APS, Denmark. Following acidification with hydrochloride acid, internal standards with deuterium labeling were administered. All samples were randomized prior to analysis. The experimental setup comprised of a high-polarity column (Zebron™ ZB-FFAP, GC Cap. Column 30 m × 0.25 mm × 0.25 μm) inserted in a gas chromatograph (7890B, Agilent) connected to a quadropole detector (5977B, Agilent) under system control of the Agilent ChemStation. The Agilent ChemStation was

further used to convert raw data into netCDF format prior to processing in Matlab R2014b (Mathworks, Inc.) using the PARADISe software as described previously[46].

**Pan-HDAC and specific HDAC activity assays.** Murine or human CD8$^+$ T cells were harvested in the lysis buffer, and 100 μl of cell lysates were subjected to HDAC inhibition by adding 5 mM of SCFAs (or 500 nM TSA) for 15 min. Following initial inhibition of HDACs by SCFAs, the peptide substrate Ac-Arg-Gly-Lys(Ac)-AMC (300 μM, Bachem, Bubendorf, Switzerland) was added to the reaction tubes for 30 min. The cleavage of the deacetylated substrate was achieved by addition of 100 μl of the developer solution (10 mg/ml trypsin in 50 mM Tris-HCl, pH = 8, 100 mM NaCl, 2 μM TSA) for 30 min at 37 °C. The fluorescence intensity of free AMC was measured at Ex/Em = 355 nm/460 nm. For the impact of bacterial supernatants and SCFAs on specific HDAC isoforms, the fluorogenic assay for each HDAC enzyme (HDAC1-3, HDAC4-6, HDAC9) was used (BPS Bioscience). Assays were performed according to the manufacturer's instructions and all measurements were conducted in triplicate using the Omega series Software V5.5, MARS 3.32 R5.

**Statistical analysis.** Means of two groups were analyzed by using an unpaired Student's *t*-test (GraphPad Prism 8). P values of $p < 0.05$ were considered significant. Where appropriate, data are presented as mean ± s.e.m. For comparison of multiple experimental groups, data were analyzed using a linear-mixed effects model with Tukey correction.

**Reporting summary.** Further information on research design is available in the Nature Research Reporting Summary linked to this article.

## Data availability

The authors declare that data supporting the findings of this study are available within the Article and its Supplementary information files. Source data are provided with this paper.

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

## Acknowledgements

We would like to thank Anne Hellhund for excellent technical support and research group A. Beilhack for providing us with OT-I mice. The technical expertise in breeding and maintaining of SPF and GF animals by staff of animal facility, Biomedical Research Center, Philipps-University of Marburg, is gratefully acknowledged. We also thank Suchita Panda, Christoph Mörtelmaier, and Iain Robertson from 4D Pharma Research Ltd. for their technical support. We are grateful to Michael Lohoff (Philipps-University of Marburg), Julia Benzel (DKFZ Heidelberg), Felix Schneider (Philipps-University of Marburg), Wolfgang Bywalez and Andrew Kaiser (Miltenyi Biotec GmbH) for helpful discussions. This project is supported by the Von Behring-Röntgen-Stiftung (Maik Luu and Ulrich Steinhoff), Stiftung PE Kempkes (Maik Luu), German Cancer Aid (Deutsche Krebshilfe e. V., Max Eder Program, grant no. 70110313 to Michael Hudecek), FAZIT-Stiftung (Hanna Leister and Alexander Visekruna) and the German Research Foundation (grants DFG-KFO325 to Thomas M Gress, Matthias Lauth, Christian A Bauer, Felix Picard, Magdalena Huber and Alexander Visekruna, DFG SFB1292 TP01 to Tobias Bopp, as well as DFG SFB/TRR 221, project no. 324392634 to Michael Hudecek).

## Author contributions

A.V., U.S., M. Hudecek, I.E.M., T.B., and T.M.G. designed and planned the study. M. Hudecek, T.N., Z.R., M. Luu, and A.V. wrote the article. M. Luu, N.R., S. Y., M.K., A.W., H.L., H.R., K.M., K.O., and C.A.B. performed in vitro experiments and analyzed the data. M. Luu, Z.R., R.R., F.P., F.F., M. Huber, and M. Lauth designed and carried out in vivo experiments. T.N and A. Baldrich performed the experiments with human CAR T cells. A. Busetti conducted the GC-MS analysis. Z.R. and A. Baldrich contributed equally to the manuscript.

## Funding

## Competing interests

M.L., M. Hudecek, and A.V. are inventors on a patent application related to the use of pentanoate that has been filed by Philipps-University Marburg and Julius-Maximilians University Würzburg (WO2021/058811A1). The title of the patent application is the following one: "Short-chain fatty acid pentanoate as enhancer for cellular therapy and anti-tumor therapy". All other authors have no competing interests.
