## [Peer Review File · Nature Communications]

Reviewers' comments:

Reviewer #1 (Remarks to the Author):

I appreciate the authors' effort to address the concerns I have raised. Among the most important ones, were whether treating mice with pentanoate in drinking water would improve adoptive T cell transfer therapy, spontaneous tumor growth or other treatment modalities that depend on CD8 T cell function such as ICB, and whether *Megasphaera massiliensis* administered by oral gavage would have a similar effect. The authors have now clearly stated in their response that only in vitro treatment of CD8 T cells with pentanoate prior transfer provides a beneficial effect. No data was provided regarding administration of *Megasphaera massiliensis*. It seems that the relevance of the present study is the potential use of in vitro pre-treatment of CD8 T cells / CAR T cells with HDACi to enhance their effector function and therapy efficacy. It is unclear what is the advantage of using a rare gut microbial metabolite to achieve this. I would suggest performing a side-by-side comparison of different HDACi including SCFAs (e.g. pentanoate and butyrate) both with human CTLs and using mouse models to validate efficacy and provide some rationale for selecting one over the other.

The authors should explicitly state that the beneficial effect of pentanoate is only observed when the cells are pre-treated in vitro. They should indicate that in vivo administration does not improve adoptive T cell transfer therapy efficacy or spontaneous tumor growth control. It is unclear from their response what happened with their attempt to monocolonize GF animals with *Megasphaera massiliensis* (a fairly commonly used approach in the field). Nevertheless, if their assumption is that it will not improve adoptive T cell transfer as they emphasize the effect is only observed in vitro, they should discuss it in the text. There is a lot of emphasis on the suggestion that *Megasphaera massiliensis* or its metabolites (e.g. pentanoate) would improve anti-tumor immunity throughout the manuscript. That is misleading. Concluding sentences in the abstract (lines 56-58), introduction (lines 103-105) and discussion (lines 383-384) suggest the administration of *Megasphaera massiliensis* or pentanoate as a potential biotherapeutics for human cancers. However, the data provided, and the authors statement in the response to the reviewers, indicate the potential is only for in vitro use. It is important to clearly state that the beneficial effect of pentanoate is only observed when T cells are treated in vitro and that in vivo administration (at least under the conditions tested by the authors) has either no effect or a detrimental one. The authors have provided new data in Extended Data 1 showing the titration of the different SCFAs tested. From that data it is clear that butyrate has a more potent immunomodulatory effect as compared to pentanoate. At the same concentration (0.5mM) they show similar toxicity levels, however, butyrate has a stronger immunomodulatory effect (as evidenced by TNF and IFN γ induction). The rationale for the use of pentanoate for the remaining of the study is that its effect on CD8 T cells (they authors had already studied its effect on other lymphocyte populations) has not been characterized. It is unclear what is the advantage of using pentanoate when one needs a much higher concentration to achieve the same effect. The authors also acknowledge that the effect of *Megasphaera massiliensis* supernatant cannot be exclusively attributed to pentanoate given that *Megasphaera massiliensis* also produces high amounts of butyrate. This should be clearly stated in the text. Ideally, butyrate, pentanoate and the supernatant should be tested in parallel.

Fig 5: A much higher concentration of pentanoate seems to be needed for human CTLs. What about toxicity? There are no statistics provided for most of Fig 5. The manuscript would benefit from more mechanistic insights obtained with human CTLs. They should include the data regarding human CTL metabolic changes, HDACi activity etc.

The authors haven't really addressed whether pentanoate increases the memory potential of CD8 T cells. From their response, it seems they used the GPR41/43 KO and assessed cytokine

production by in vitro generated CTLs. They have not measured memory features (CD44 expression alone wouldn't necessarily imply memory) or assessed their memory potential after recall responses. It is also possible that the effect of pentanoate is GPR41/43- independent.

Line 100: The authors should state in vitro treatment with the particular SCFAs or supernatant of *Megasphaera massiliensis*.

In Fig 2 for Reviewer 1 the authors used the B16 model to assess the effect of systemic administration of pentanoate in anti-PD1 treatment. From this data the authors concluded that pentanoate only works in the adoptive transfer model with Ag-specific T cells. However, they have not extended their findings to a model other than OVA/OT-I. Other models are available and could be tested to generalize their findings (e.g. pmel TCR transgenic mice, specific for the melanoma Ag gp100).

Statistical analysis still lacks correction for multiple comparisons in most figures where more than two groups/conditions are compared (it was only done for Fig 3b,h). This should be performed and clearly stated in the legend and methods. Statistical analysis for Fig 2g, tumor growth curve, should also consider repeated measures. The majority of the experiments still contain very few biological replicates (2-3) per experiment and have been done only twice. The authors indicate that they attempted to overcome the small sample size but only provide combined data from 2 independent experiments (n=3/experiment) in Fig 3.

Some legends are still unclear. E.g. Fig 4g: "n=3, performed in 3 independent experiments" Does this mean one well per experiment? No technical replicates? Same for 4i-j, were there replicates in each experiment? Fig 2d-e, the legend states 3 independent experiments were performed but it is unclear what is shown in e. One representative experiment or each dot represents one experiment?

New figure Extended Data 3c: Please provide quantification and statistical analysis for glycolysis, glycolytic capacity and glycolytic reserve.

The authors acknowledge that the staining necessary to determine whether pentanoate-treated CD8 T cells do indeed expand is not difficult to perform. Unfortunately, however, they have not done the experiment and no new data is provided to confirm that the cells expand. They have now included absolute numbers and they speculate in the response to the reviewer that the increased number is probably due to both expansion and survival. The title of the section in the manuscript states "Pentanoate promotes the expression of CD25 and IL-2 as well as in vivo expansion and persistence of CTLs", providing at least data that shows that the cells proliferate would support that statement. Adding further mechanistic insight, would certainly strengthen this manuscript.

Line 102: Longevity of CTLs is not actually shown.

Line 115-116 and Fig 1a: The authors have already shown very similar data in their previous study (Luu et al). Could use the space for other data if needed.

Line 124: The effect of other products present in the luminal content of colon and caecum cannot be ruled out. Please rephrase.

Line 786 correct "CLTs"

Extended Data Fig. 3a only shows one representative flow cytometry plot. Please provide quantification of the data and corresponding statistical analysis.

Extended Data 4: Provide statistical analysis for 4a. Clarify in legend how many mice per group in each experiment for all the panels.

Line 151: The authors should provide a rationale for the analysis of HDAC inhibitory function and acknowledge what is already known regarding HDACi activity of SCFAs. It should be stated that the HDAC inhibitory function of pentanoate has been previously shown. The Class I HDAC

inhibitory activity of *Megasphaera massiliensis* supernatant and pentanoate previously described by Yuille S et al, 2018 should be mentioned here. The way it is currently stated comes across as a new finding.

Fig 2h-i: Indicate in what tissue is shown.

Fig 3e: How were the quadrants set? In general, flow plots should include the control used to set the gates for positive/negative populations.

Fig 3d: Please indicate time point.

Line 243-245: The authors state "we detected a higher number of pentanoate-treated cells expressing more TNF- α and IFN- γ in comparison to control OT-I CTLs in tumors, draining LNs and spleen", however, there is no difference in the freq of IFN γ + TNF+ in iLN and numbers are only provided for tumor. Please correct.

Reviewer #3 (Remarks to the Author):

The authors demonstrate that pentanoate, possibly through HDACi mechanism leads to enhanced effector activity and possibly memory (although this is not formally demonstrated).

This is a resubmission and response to several of the original comments are addressed.

1. in vitro data demonstrate that in murine CD8 T cells exposure to *M. Massiliensis* SN (and pentanoate), leads to enhanced number and function of CTL; in vivo models using B16OVA and PancOVA demonstrate reduction in tumor volume in compared with those receiving CTL not exposed to pentanoate.

- As already critiqued in the original review, these two models using a xenoantigen are not sufficiently rigorous to establish this finding as a general phenomenon, especially since several readily available syngeneic models are available, e.g pmel/ B16 that targets an endogenous antigen target. or even, MC38, a more immunogenic model would have been acceptable.

2. The authors claim that this would be effective in CAR-T cells, yet this was not demonstrated even with in vitro data

3. The absence of human antigen-specific CTL for any of the in vitro studies to demonstrate potential translatability is troubling

4. As the authors state, the physiological amount of pentanoate is low which is the reason for changing the focus of the study, acknowledging also the study published by Yuille et al describing the HDACi activity of *M. Massiliensis*.

In essence then, the main findings of the manuscript have been reduced to the finding that pentanoate at supraphysiological doses, ie. in vitro, can have a HDACi -like effect on a limited set of mouse T cells, altho the mechanism of action as the authors admit is not well defined.

Point-by-point response

Reviewer #1 (Remarks to the Author):

I would suggest performing a side-by-side comparison of different HDACi including SCFAs (e.g. pentanoate and butyrate) both with human CTLs and using mouse models to validate efficacy and provide some rationale for selecting one over the other.

As suggested by this reviewer, we have performed additional experiments. The novel data suggest that pentanoate and butyrate act via integrated metabolic and epigenetic reprogramming of cells. Both SCFAs acted on CTLs and CAR T cells through inhibition of class I histone deacetylase (HDAC) activity and by enhancing the function of mTOR. In contrast to SCFAs, the chemical inhibitor of class I HDAC isoforms (mocetinostat) was not capable of altering the metabolic status and mTOR activity on CTLs. In vivo effect of SCFA-treated CTLs was more pronounced than that of mocetinostat-treated CTLs. The class II HDAC inhibitor TMP-195 did not show any significant effect on CTLs.

The authors should explicitly state that the beneficial effect of pentanoate is only observed when the cells are pre-treated in vitro. They should indicate that in vivo administration does not improve adoptive T cell transfer therapy efficacy or spontaneous tumor growth control. It is unclear from their response what happened with their attempt to monocolonize GF animals with *Megasphaera massiliensis* (a fairly commonly used approach in the field). Nevertheless, if their assumption is that it will not improve adoptive T cell transfer as they emphasize the effect is only observed in vitro, they should discuss it in the text. There is a lot of emphasis on the suggestion that *Megasphaera massiliensis* or its metabolites (e.g. pentanoate) would improve anti-tumor immunity throughout the manuscript. That is misleading. Concluding sentences in the abstract (lines 56-58), introduction (lines 103-105) and discussion (lines 383-384) suggest the administration of *Megasphaera massiliensis* or pentanoate as a potential biotherapeutics for human cancers. However, the data provided, and the authors statement in the response to the reviewers, indicate the potential is only for in vitro use. It is important to clearly state that the beneficial effect of pentanoate is only observed when T cells are treated in vitro and that in vivo administration (at least under the conditions tested by the authors) has either no effect or a detrimental one.

We have removed all misleading statements as well as concluding sentences (lines 56-58, 103-105 and 383-384) in the revised manuscript following the reviewer's advice. We now clearly state that there is only the therapeutic potential for *in vitro* use of two SCFAs, butyrate and pentanoate. We also show that *in vivo* administration of pentanoate does not improve anti-tumor immune responses and has no beneficial effect on anti-PD-1 therapy (novel Extended Data Fig. 7).

The authors have provided new data in Extended Data 1 showing the titration of the different SCFAs tested. From that data it is clear that butyrate has a more potent immunomodulatory

effect as compared to pentanoate. At the same concentration (0.5mM) they show similar toxicity levels, however, butyrate has a stronger immunomodulatory effect (as evidenced by TNF and IFN γ induction). The rationale for the use of pentanoate for the remaining of the study is that its effect on CD8 T cells (they authors had already studied its effect on other lymphocyte populations) has not been characterized. It is unclear what is the advantage of using pentanoate when one needs a much higher concentration to achieve the same effect.

We appreciate the concern of the reviewer and have duly addressed the concern by repeating many of the key experiments. We agree that there is no advantage of using pentaonate over butyrate. In contrast, both molecule might be tested for a therapeutic approach and butyrate seems to be even more potent than pentanoate. We removed all misleading statements from the manuscript. We now state that both butyrate and pentanoate might be therapeutically interesting molecules with a potential for cell-based cancer immunotherapy. We thank the reviewer for this critical point.

The authors also acknowledge that the effect of *Megasphaera massiliensis* supernatant cannot be exclusively attributed to pentanoate given that *Megasphaera massiliensis* also produces high amounts of butyrate. This should be clearly stated in the text. Ideally, butyrate, pentanoate and the supernatant should be tested in parallel.

We have performed additional parallel experiments as suggested by the reviewer and found that both, butyrate and pentanoate (as well as the supernatant of *M. massiliensis*) promote beneficial effects. We have also changed the text following the reviewer's advice.

Fig 5: A much higher concentration of pentanoate seems to be needed for human CTLs. What about toxicity? There are no statistics provided for most of Fig 5. The manuscript would benefit from more mechanistic insights obtained with human CTLs. They should include the data regarding human CTL metabolic changes, HDACi activity etc.

We have routinely checked for the toxicity of pentanoate and other compounds used in this study. We appreciate the concern of the reviewer and have now included the data with regard to human CTL metabolic changes and HDACi activity into the manuscript (Extended Data Fig 9). We also included statistical analysis into the Fig 5.

The authors haven't really addressed whether pentanoate increases the memory potential of CD8 T cells. From their response, it seems they used the GPR41/43 KO and assessed cytokine production by in vitro generated CTLs. They have not measured memory features (CD44 expression alone wouldn't necessary imply memory) or assessed their memory potential after recall responses. It is also possible that the effect of pentanoate is GPR41/43- independent.

We appreciate the suggestion of the reviewer however we think that this issue is beyond the scope of this manuscript.

Line 100: The authors should state in vitro treatment with the particular SCFAs or supernatant of *Megasphaera massiliensis*.#

We have changed the text according to the reviewer's advice.

In Fig 2 for Reviewer 1 the authors used the B16 model to assess the effect of systemic administration of pentanoate in anti-PD1 treatment. From this data the authors concluded that pentanoate only works in the adoptive transfer model with Ag-specific T cells. However, they have not extended their findings to a model other than OVA/OT-I. Other models are available and could be tested to generalize their findings (e.g. pmel TCR transgenic mice, specific for the melanoma Ag gp100).

In order to generalize our finding for both CTLs and CAR T cells, we included novel Figure in the manuscript (novel Fig. 4). Based on the obtained data, we conclude that the observed beneficial effect can be applied to other tumor-derived antigens that are more physiological as shown in the novel Figure 4e.

Statistical analysis still lacks correction for multiple comparisons in most figures where more than two groups/conditions are compared (it was only done for Fig 3b,h). This should be performed and clearly stated in the legend and methods. Statistical analysis for Fig 2g, tumor growth curve, should also consider repeated measures. The majority of the experiments still contain very few biological replicates (2-3) per experiment and have been done only twice. The authors indicate that they attempted to overcome the small sample size but only provide combined data from 2 independent experiments ($n=3/\text{experiment}$) in Fig 3.

We appreciate the concern of the reviewer and have duly addressed the concern by including more biological replicates and experiments (please see the novel Fig. 3 and Fig. 4).

Some legends are still unclear. E.g. Fig 4g: "n=3, performed in 3 independent experiments" Does this mean one well per experiment? No technical replicates? Same for 4i-j, were there replicates in each experiment? Fig 2d-e, the legend states 3 independent experiments were performed but it is unclear what is shown in e. One representative experiment or each dot represents one experiment?

We thank the reviewer for this comment. We have improved the figure legend of these figures.

New figure Extended Data 3c: Please provide quantification and statistical analysis for glycolysis, glycolytic capacity and glycolytic reserve.

We have included the quantification and statistics for glycolysis, glycolytic capacity and glycolytic reserve into the figure.

The authors acknowledge that the staining necessary to determine whether pentanoate-treated CD8 T cells do indeed expand is not difficult to perform. Unfortunately, however, they have not done the experiment and no new data is provided to confirm that the cells expand. They have now included absolute numbers and they speculate in the response to the reviewer that the increased number is probably due to both expansion and survival. The title of the section

in the manuscript states “Pentanoate promotes the expression of CD25 and IL-2 as well as in vivo expansion and persistence of CTLs”, providing at least data that shows that the cells proliferate would support that statement. Adding further mechanistic insight, would certainly strengthen this manuscript.

We thank the reviewer for pointing this out: As suggested by the reviewer, we performed additional experiments and found that pentanoate-pretreated and CFSE-labeled CTLs had a stronger proliferative capacity in vivo (as compared to control CTLs). We have included these results into the manuscript (Extended Data Fig. 8e).

Line 102: Longevity of CTLs is not actually shown.

We have removed this statement from the text according to the reviewer’s advice.

Line 115-116 and Fig 1a: The authors have already shown very similar data in their previous study (Luu et al). Could use the space for other data if needed.

We thank the reviewer for this suggestion.

Line 124: The effect of other products present in the luminal content of colon and caecum cannot be ruled out. Please rephrase.

We thank the reviewer for pointing this out. We have included an additional sentences stating that the effect of other microbial products cannot be ruled out.

Line 786 correct “CLTs”

We have changed the text.

Extended Data Fig. 3a only shows one representative flow cytometry plot. Please provide quantification of the data and corresponding statistical analysis.

We appreciate the concern of the reviewer and have now addressed this point.

Extended Data 4: Provide statistical analysis for 4a. Clarify in legend how many mice per group in each experiment for all the panels.

We thank the referee for this comment. We have changed the text.

Line 151: The authors should provide a rationale for the analysis of HDAC inhibitory function and acknowledge what is already known regarding HDACi activity of SCFAs. It should be stated that the HDAC inhibitory function of pentanoate has been previously shown. The Class I HDAC inhibitory activity of *Megasphaera massiliensis* supernatant and pentanoate previously described by Yuille S et al, 2018 should be mentioned here. The way it is currently stated comes across as a new finding.

We have now several times indicated that the HDAC inhibitory activity of pentanoate has been previously shown by Samantha Yuille and colleagues, who (together with her group leader

Imke E Mulder and two other scientists from Aberdeen, UK) is the co-author of the current study as well.

Fig 2h-i: Indicate in what tissue is shown.

We have now indicated in the text the tissue shown in the figure.

Fig 3e: How were the quadrants set? In general, flow plots should include the control used to set the gates for positive/negative populations.

We have now included the control used to set the gates as well as the gating strategy in the Extended Data Fig. 10.

Fig 3d: Please indicate time point.

We have now indicated the time point in the figure legend.

Line 243-245: The authors state “we detected a higher number of pentanoate-treated cells expressing more TNF- α and IFN- γ in comparison to control OT-I CTLs in tumors, draining LNs and spleen”, however, there is no difference in the freq of IFN γ ⁺ TNF⁺ in iLN and numbers are only provided for tumor. Please correct.

We thank the reviewer for this important issue. We have now updated the Fig 3.

Reviewer #3 (Remarks to the Author):

The authors demonstrate that pentanoate, possibly through HDACi mechanism leads to enhanced effector activity and possibly memory (although this is not formally demonstrated).

This is a resubmission and response to several of the original comments are addressed.

1. *in vitro* data demonstrate that in murine CD8 T cells exposure to *M. Massiliensis* SN (and pentanoate), leads to enhanced number and function of CTL; *in vivo* models using B16OVA and PancOVA demonstrate reduction in tumor volume in compared with those receiving CTL not exposed to pentanoate.

- As already critiqued in the original review, these two models using a xenoantigen are not sufficiently rigorous to establish this finding as a general phenomenon, especially since several readily available syngeneic models are available, e..g pmel/ B16 that targets an endogenous antigen target. or even, MC38, a more immunogenic model would have been acceptable.

We understand the concern of the reviewer regarding this point. In the novel Fig. 4, we have performed experiments with physiologically relevant protein ROR1, which is expressed on many epithelial tumors such as pancreatic cancer and lung cancer, and murine CAR T cells. The experiments in this cancer model show similar results to already shown data sets.

2. The authors claim that this would be effective in CAR-T cells, yet this was not demonstrated even with *in vitro* data

We thank the review for this important comment. In the Fig. 5g (which is now the Fig. 5i in the novel version of the manuscript), we have already shown that human pentanoate-treated CAR T cells have a superior cytolytic activity over control CAR T cells (*in vitro* T cell killing assay). Moreover, in the novel Fig. 4e, we now demonstrate that murine pentanoate-treated CAR T cells enhanced anti-tumor immunity of CAR T cells (*in vivo* model).

3. The absence of human antigen-specific CTL for any of the *in vitro* studios to demonstrate potential translatability is troubling

We understand the concern of the reviewer regarding the data with antigen-specific T cells. For several years, our group (Michael Hudecek lab) has been involved into efficient generation of human antigen-specific CD8⁺ CAR T cells. In the Fig. 5, the reviewer can find the data generated with antigen-specific human CAR T cells. In addition, in the novel Fig. 4, we now show results generated with murine antigen-specific CAR T cells.

4. As the authors state, the physiological amount of penatanoate is low which is the reason for changing the focus of the study, acknowledging also the study published by Yuille et al describing the HDACi activity of *M. Massiliensis*.

In essence then, the main findings of the manuscript have been reduced to the finding that pentanoate at supraphysiolgoic doses, ie. *in vitro*, can have a HDACi -like effect on a limited set of mouse T cells, altho the mechanism of action as the authors admit is not well defined.

We appreciate these important suggestions and comments of the reviewer and have been investigating the role of SCFAs in much more details. We have performed several additional experiments with antigen-specific CAR T cells and murine and human CTLs. We clearly state that the idea of this study is to exploit the therapeutic potential of physiologically abundant substances such as pentanoate and butyrate for T cell-based cancer therapies. The mechanism of action is clearly defined by us: We show that SCFAs pentanoate and butyrate reprogram histone deacetylase (HDAC) and additionally modulate mTOR activity in CD8⁺ T cells, which is the advantage of these molecules over other HDAC inhibitors (Please see the novel Fig. 3b-g). We also show that antigen-specific SCFA-treated CAR T cells are capable of persisting in the immunosuppressive environment of solid tumors *in vivo*, which we think is a novel and very important translational aspect. To our knowledge, our current study is the first paper demonstrating that microbiota-derived SCFAs are able to enhance the CAR T cell- and CTL-mediated immunotherapy. These data clearly indicate the therapeutic potential for the SCFAs pentanoate and butyrate.

We would like to thank the reviewer for this thoughtful review. We truly hope that these changes to the manuscript will facilitate the decision made by the reviewer.

Reviewers' comments:

Reviewer #1 (Remarks to the Author):

While I appreciate the authors' effort to address the reviewers' concerns, I believe the major concerns have not been fully addressed. Based on their title, abstract, introduction, and discussion the authors continue to frame their study as an important advancement in our understanding of how microbiota regulates anti-tumor immunity. However, the main finding of the paper is the potential use of in vitro pre-treatment of CD8 T cells / CAR T cells with butyrate or pentanoate to enhance adoptive cell transfer efficacy. In terms of the mechanistic understanding, both the HDACi activity and impact on T cell metabolism of the two SCFAs used in the study have been shown before, raising concerns regarding the novelty of the manuscript. The authors could focus on the in vitro potential of the compounds to improve adoptive cell transfer efficacy and provide additional data regarding T cell function/mechanism. Given that the authors have clearly established that the benefit of the SCFAs is only when cells are pre-treated in vitro, the microbiota related data in Figures 1 and 2 is not really necessary. From a translational perspective, it wouldn't be practical to isolate the particular commensals to generate supernatant to use for in vitro treatment of T cells when it does not provide any additional benefit over the purified SCFAs.

Although the authors provided new data comparing butyrate and pentanoate to other HDAC inhibitors as suggested, this was only partially done.

In the point-by-point response to reviewers, the authors indicate that they routinely checked for toxicity of pentanoate and other compounds used. However, the data is not provided for the experiments with human cells.

The authors still only use 3 replicates per experiment even for in vivo studies. The new figures they mention in the response letter show two experiments with n=3 each. For in vitro experiments it's still unclear what the dots represent "3 dots shown and legend states n=3 combined from 3 independent experiments"

My concerns regarding statistical analyses have not been properly addressed. The majority of the figures include comparison of several groups/conditions, yet most legends state that two-tailed unpaired Student's t-test was performed but no correction for multiple comparisons indicated. In many figures it seems that the authors only selected some comparisons to indicate significance. Were the others not significant or not compared?

In figures 3 and 4, the authors indicated in the legend that "Multiple group comparison was performed by a linear-mixed effects model with Tukey correction". I'm assuming they refer to panels b and h for Fig 3 (although not clearly stated) and they refer to panel e for Fig 4. However, those figures still show the statistical significance (i.e. *, **, ns) for individual time points. This is concerning given that a linear-mixed effects model would normally give a p value for the whole curve rather than for each time point. Please clarify. Also indicate in the methods whether interactions were tested.

In the methods they included the following statement: "For comparison of multiple experimental groups, data were analyzed using a linear-mixed effects model with Tukey correction." Is that what they've done for all the figures? Applied a linear-mixed effects model? In a study like this one, it would usually be applied when multiple measurements over time are taken (e.g. tumor growth curve). Please clarify.

Fig. 1a-d: The comparison should be against untreated and not between SPF and GF (and corrected).

Line 133: should mention SCFAs and other bacterial products

Line 136: the rationale to bring this up here is not clear, I suggest changing the order with line 140.

Line 140: Authors should cite here the work that has already shown that SCFAs, particularly butyrate, alter metabolic status and function of CD8 T cells. Even their own work with pentanoate

and CD4 T cells.

As indicated above, the section starting in line 179 does not seem particularly relevant for the in vitro effect.

Line 212-214: Please cite figure.

Figure 3b-g: Please indicate for panel e whether that is percentage of CD8, CD45 or total cells? Explain the tSNE. What is it showing? Is that LN? What is it gated on? What marker is shown? The title on top says CD8CD45, is that the gate or was the tSNE used to gate on those cells? Were all LN combined and run together and each quadrant represents the different groups/treatment conditions?

For CFSE experiments, rather than MFI the authors should show proliferation index and precursor frequency. The authors should include earlier time points after transfer to show the proliferating treated cells.

Extended data fig 8j: What are the culture conditions? Data regarding cell viability and IL-2 consumption would support their conclusion.

Line 317: That statement is incorrect, there is no data that *A. muciniphila* restores ICI efficacy in nonresponsive patients. Please re-phrase

Reviewer #3: report not provided

Reviewer #4 (Remarks to the Author): to replace #3

Maik Luu and colleagues demonstrate that the SCFA pentanoate, derived from the non-frequent commensal *M. massiliensis*, impacts persistence and anti-tumor activity of CD8 T cells.

My major suggestion is to add data, or at least extend the discussion regarding the potential mechanism of pentanoate-induced CD8 T cell activity. Looking at the presented in vitro and in vivo data, there are 3 lines of data: (1) up-regulated expression CD25; (2) enhanced glycolysis; and (3) down-regulated activity of HDAC class I. The question becomes whether these effects are inter-related or inter-dependent or not. In other words, authors should consider to intervene with pentanoate-stimulation of CD8 T cells using reagents to block CD25, mTOR and/or HDAC. In the current manuscript, this has only been done with HDAC inhibitors. Alternatively, authors should consider whole genome expression data and pathway analysis upon pentanoate stimulation of T cells. Such data may enable assessment of hierarchical dependency and better point to mechanism of action. For instance, is up-regulated expression of CD25 a prerequisite for the mTOR activity and/or HDAC inhibition, and which of these effects contributes most to CD8 T cell IFN γ production. In addition, which molecular characteristics are likely to make pentanoate particularly active compared to other SCFAs; i.e., the use of certain GPRs. Data or discussion towards MOA is recommended.

Minor suggestions are as follows:

- provide details on human CAR construct, where is tEGFR put into vector.
- add cartoons regarding T cell stimulation (anti-CD3/28 and IL2) in figures 1 and 3.
- why is butyrate not included in Figs 3H/4E; and why have authors (in Figs 1-5) not consistently shown IFN γ +, TNF α + CD8 T cells (rather than sometimes one, sometimes the other, sometimes both cytokines).
- discuss metabolic and/or epigenetic requirements for pentanoate sensitivity; for instance, are T cells that express immune checkpoints potentially less sensitive for pentanoate.
- discuss why pentanoate-stimulation of T cells is preferred over in vivo administration of pentanoate; also discuss the relevance of providing *M. massiliensis* as a probiotic for patients receiving immune therapy.

Point-by-point response

Reviewer #1 (Remarks to the Author):

While I appreciate the authors' effort to address the reviewers' concerns, I believe the major concerns have not been fully addressed. Based on their title, abstract, introduction, and discussion the authors continue to frame their study as an important advancement in our understanding of how microbiota regulates anti-tumor immunity. However, the main finding of the paper is the potential use of in vitro pre-treatment of CD8 T cells / CAR T cells with butyrate or pentanoate to enhance adoptive cell transfer efficacy. In terms of the mechanistic understanding, both the HDACi activity and impact on T cell metabolism of the two SCFAs used in the study have been shown before, raising concerns regarding the novelty of the manuscript. The authors could focus on the in vitro potential of the compounds to improve adoptive cell transfer efficacy and provide additional data regarding T cell function/mechanism. Given that the authors have clearly established that the benefit of the SCFAs

is only when cells are pre-treated in vitro, the microbiota related data in Figures 1 and 2 is not really necessary. From a translational perspective, it wouldn't be practical to isolate the particular commensals to generate supernatant to use for in vitro treatment of T cells when it does not provide any additional benefit over the purified SCFAs. Although the authors

provided new data comparing butyrate and pentanoate to other HDAC inhibitors as suggested, this was only partially done.

Author response: We thank the reviewer for this essential point. As requested by the reviewer, we have moved the focus of the manuscript from the microbiome to SCFAs. We now entirely focus on the potential of SCFAs to improve the adoptive cell immunotherapy for cancer (through *ex vivo* treatment of CTLs with these SCFAs). We have removed microbiota-related data from the Fig 1a-e, and Suppl. Figure 4 (Extended Data Fig. 4) from the manuscript. The microbiota-related results from the former Fig. 2b are now in the Supplement (Extended Data Fig. 1). We only kept the screening data from the LC-MS in order to introduce small compounds (SCFAs) and their potential to inhibit HDAC activity (Novel Fig. 1a). We hope that the Reviewer can accept that these results serve as a way of introducing the SCFAs analyzed in the manuscript.

In the point-by-point response to reviewers, the authors indicate that they routinely checked for toxicity of pentanoate and other compounds used. However, the data is not provided for the experiments with human cells.

Author response: We appreciate the concern of the reviewer and have now added the missing data into the Supplement (Extended Data Fig. 9a).

The authors still only use 3 replicates per experiment even for *in vivo* studies. The new figures they mention in the response letter show two experiments with $n=3$ each. For *in vitro* experiments it's still unclear what the dots represent "3 dots shown and legend states $n=3$ combined from 3 independent experiments"

Author response: As shown in the respective figures, 6 replicates have been used for *in vivo* studies. In *in vitro* studies, each dot represents an individual experiment (an individual mice). Data from three independent experiments have been pooled.

My concerns regarding statistical analyses have not been properly addressed. The majority of the figures include comparison of several groups/conditions, yet most legends state that two-tailed unpaired Student's t-test was performed but no correction for multiple comparisons indicated. In many figures it seems that the authors only selected some comparisons to indicate significance. Were the others not significant or not compared?

Author response: All statistical analyses have been thoroughly reanalyzed and the clarification is now provided. Multiple comparisons have only been performed for curves

derived from in vivo studies. In diagrams, only two groups were compared with each other as indicated in the figures. The other groups were not compared.

In figures 3 and 4, the authors indicated in the legend that “Multiple group comparison was performed by a linear-mixed effects model with Tukey correction”. I’m assuming the refer to panels b and h for Fig 3 (although not clearly stated) and they refer to panel e for Fig 4. However, those figures still show the statistical significance (i.e. *, **, ns) for individual time points. This is concerning given that a linear-mixed effects model would normally give a p value for the whole curve rather than for each time point. Please clarify. Also indicate in the methods whether interactions were tested.

Author response: As suggested by the reviewer we have re- analyzed the data comparing the whole curves with each other using the linear-mixed effects model. We have changed the Fig. 3 b, h and Fig. 4 e accordingly. We further included clarification in the figure legend.

In the methods they included the following statement: “For comparison of multiple experimental groups, data were analyzed using a linear-mixed effects model with Tukey correction.” Is that what they’ve done for all the figures? Applied a linear-mixed effects model? In a study like this one, it would usually be applied when multiple measurements over time are taken (e.g. tumor growth curve). Please clarify.

Author response: The linear-mixed analysis was only performed for multiple comparisons in the context of tumor growth curves (Fig. 3 b, h and Fig. 4 e). For comparison of two groups, the student t-test was performed between the indicated conditions. We have clarified this in the methods explicitly now.

Fig. 1a-d: The comparison should be against untreated and not between SPF and GF (and corrected).

Author response: We agree with the Reviewer. The sample upper left is an untreated control, which we had not assigned previously. As mentioned before, these data have been removed from the manuscript, as suggested by the reviewer.

Line 133: should mention SCFAs and other bacterial products

Author response: As suggested by the Reviewer (“the microbiota related data in Figures 1 and 2 is not really necessary.”), we removed the microbiota-related data and corresponding text (also line 133) from the manuscript.

Line 136: the rationale to bring this up here is not clear, I suggest changing the order with line 140.

Author response: We agree with the reviewer and have revised the text as suggested.

Line 140: Authors should cite here the work that has already shown that SCFAs, particularly butyrate, alter metabolic status and function of CD8 T cells. Even their own work with pentanoate and CD4 T cells.

Author response: We thank the reviewer for this comment. We have now cited the relevant work.

As indicated above, the section starting in line 179 does not seem particularly relevant for the in vitro effect.

Author response: We thank the reviewer. We have removed the main part of this section from the manuscript.

Line 212-214: Please cite figure.

Author response: We have now cited the figure.

Figure 3b-g: Please indicate for panel e whether that is percentage of CD8, CD45 or total cells?

Explain the tSNE. What is it showing? Is that LN? What is it gated on? What marker is shown? The title on top says CD8CD45, is that the gate or was the tSNE used to gate on those cells? Were all LN combined and run together and each quadrant represents the different groups/treatment conditions?

Author response: As suggested by the reviewer, we have indicated in Fig. 3 e and in the figure legend that it refers to the percentage of CD8⁺ CD45.1⁺ cells. Each t-SNE panel shows a representative, individual sample of the LN from mice of the respective group/ treatment condition. The samples were gated on lymphocytes, whereas the panels indicate the CD8⁺CD45.1⁺ population among these. We have clarified this in the figure legend now.

For CFSE experiments, rather than MFI the authors should show proliferation index and precursor frequency. The authors should include earlier time points after transfer to show the proliferating treated cells.

Author response: We thank the reviewer. We have included the proliferation index into the figure.

Extended data fig 8j: What are the culture conditions? Data regarding cell viability and IL-2 consumption would support their conclusion.

Author response: For experiments in Ext. Data Fig. 8j, CD8⁺ T cells were cultured as described in the methods with anti-CD3, anti-CD28 and rhIL-2 for activation. Supernatant was collected on the indicated days post activation and analyzed for murine IL-2. We ensured sufficient cell viability by analysis of the cells as shown in Ext. Data Fig. 2a.

Line 317: That statement is incorrect, there is no data that *A. muciniphila* restores ICI efficacy in nonresponsive patients. Please re-phrase.

Author response: We apologize for this oversight and the incorrect statement. We have changed the text accordingly.

Reviewer #4 (Remarks to the Author): to replace #3

Maik Luu and colleagues demonstrate that the SCFA pentanoate, derived from the non-frequent commensal *M. massiliensis*, impacts persistence and anti-tumor activity of CD8 T cells.

My major suggestion is to add data, or at least extend the discussion regarding the potential mechanism of pentanoate-induced CD8 T cell activity. Looking at the presented in vitro and in vivo data, there are 3 lines of data: up-regulated expression CD25; (2) enhanced glycolysis; and (3) down-regulated activity of HDAC class I. The question becomes whether these effects are inter-related or inter-dependent or not. In other words, authors should consider to intervene with pentanoate-stimulation of CD8 T cells using reagents to block CD25, mTOR and/or HDAC. In the current manuscript, this has only been done with HDAC inhibitors. Alternatively, authors should consider whole genome expression data and pathway analysis upon pentanoate stimulation of T cells. Such data may enable assessment of hierarchical dependency and better point to mechanism of action. For instance, is up-regulated expression of CD25 a prerequisite for the mTOR activity and/or HDAC inhibition, and which of these effects contributes most to CD8 T cell IFN γ production. In addition, which molecular characteristics are likely to make pentanoate particularly active compared to other SCFAs;

i.e., the use of certain GPRs. Data or discussion towards MOA is recommended.

Author response: We thank the reviewer for these important comments and suggestions that certainly will improve the quality of this manuscript. According to the literature, only “5–10% of gene transcription has been shown to be influenced by HDAC inhibitor treatment, with approximately half of the affected genes being upregulated and half being downregulated” (Hull E.E. *et al.*, Biomed Res Int. 2016). Our novel experimental data demonstrate that butyrate and pentanoate treatment of CD8⁺ T cells increase acetylation of histones H3 and H4 (Reviewer-only Figure 1a). Furthermore, the HDAC-inhibitory effect of pentanoate on CTLs was investigated by ChIP assay for proximal promoter regions of *Ifng*, *Eomes* and *Tbx21* genes. We have observed that pentanoate was capable of increasing H4 acetylation at the *Eomes* and *Ifng* loci, and to less extent at the *Tbx21* locus (Reviewer-only Figure 1b).

Reviewer-only Figure 1. **a**, Western blot analysis of pan-H3Ac and pan-H4Ac in CD8⁺ T cells treated with SCFAs for 3 days. One representative experiment is shown (n=3 independent experiments). **b**, ChIP qPCR analysis for acetylation of histone H4 at the promoter regions of *Ifng*, *Eomes* and *Tbx21* in pentanoate-treated CTLs. Data are from pooled chromatin of three mice and are shown as mean \pm s.d. **c**, RNA-seq analysis of CD8⁺ T cells treated with 2.5 mM pentanoate for three days. Volcano plot with differentially regulated genes is shown. **d**, Expression of CTL-associated genes in pentanoate-treated

CD4⁺ T cells. Results of RNA-seq analysis for indicated genes are displayed as reads per kilo base per million mapped reads (RPKM).

Moreover, independently of T cell subset investigated, pentanoate induced a set of genes associated with CTL function (see please our RNA-seq data for pentanoate-treated CD4⁺ T cells, Reviewer-only Figure 1c,d). Together with the Wilfried Ellmeier lab, we have recently analyzed CD4⁺ T cell with a specific deletion of the class I histone deacetylases HDAC1 and HDAC2 (pentanoate is a specific inhibitor of these both enzymes, as shown by us in the Fig. 2 of the current manuscript) and performed RNA-seq for T cells lacking both HDAC1 and HDAC2. We observed a strong increase in the expression of *Eomes*, *Tbx21*, *GzmB* *Prf1* and *Ifng* within CD4⁺ T cells in the absence of HDAC1 and HDAC2 (Preglej T. *et al.*, JCI Insight, 2020). These RNA-seq data set (Fig 2a, Preglej T. *et al.*) strongly resembles our RNA-seq data for pentanoate-treated T cells (Reviewer-only Figure 1c,d). We think that pentanoate-mediated effects on CTLs were predominantly mediated via HDAC inhibition, but we do not exclude inter-dependent (epigenetics-metabolism) effects. We have now discussed all these aspects in the discussion and we cite the corresponding literature and our previous work.

We have also tried to better understand the complex metabolic-epigenetic interactions mediated by SCFAs. Both, the cytokine production and cytolytic activity of CTLs have been shown to be dependent on glucose metabolism (Cham CM *et al.*, Eur. J. Immunol., 2008). Recently, it was demonstrated that SCFAs act as substrates for the cellular metabolism in immune cells (Park J. *et al.*, Mucosal Immunology, 2014, Balmer E.L. *et al.* Immunity 2016). On the one side, SCFAs are able to regulate the cytokine expression and function of immune cells of lymphocytes by rewiring their metabolic properties. On the other hand, we were not able to observe acetate-mediated induction of granzyme B and perforin, suggesting that some CTL-related molecules are strictly dependent on epigenetic modulation but not on metabolic alterations. Importantly, acetate is the only SCFA without any detectable HDAC inhibitory activity.

Other groups have described that not only SCFAs, but also TSA can enhance the activity of mTOR in T cells (Park J *et al.*, Mucosal immunology, 2015). We do see the effect of TSA on mTOR activity, but to less extent than that of SCFAs. Intriguingly, the HDAC class I and II specific HDAC inhibitors such as mocetinostat and TMP-195 were not able to increase the phosphorylation of mTOR and its target protein S6. There seem to be a certain inter-

dependence between HDAC inhibition and metabolic effects, which is still not well understood. We have discussed all these important aspect in the section discussion.

We have also performed additional experiments with regard to a possible role for GPR41 and GPR43 in CTLs, as suggested by the reviewer. Although SCFAs regulate the activity of various cells by to cognate G-protein-coupled receptors (GPR), we were not able to detect any dependence of SCFA-mediated effects in CTLs on GPR41 and GPR43 (mice lacking two most important SCFA-receptors, GPR41 and GPR43 were kindly provided by Prof. Stefan Offermanns, Max Planck Institute for Heart and Lung Research). Moreover, the inhibitory activity of pentanoate on HDAC was not affected in cell lysates derived from DKO mice (GPR41 and GPR43 KO), suggesting that those effects are not inter-related. Thus, our novel data indicate that the HDAC inhibition and cytokine production in CTLs (mediated by pentanoate) is independent of GPR41 and GPR43. We have included these results in the Supplementary Fig 5. We have discussed the independence of SCFA-mediated effects on GPRs in the section discussion.

Finally, as requested by the reviewer, we have performed additional experiments with regard to the pentanoate-mediated upregulation of CD25. We found that the pentanoate-triggered induction of CD25 was completely abrogated upon a co-treatment of CTLs with 2-DG. There seem to be an inter-related effect between CD25 upregulation, pentanoate-mediated enhanced glycolysis and mTOR activity. IL-2 is known to induce mTOR via CD25 in T cells, which in turns enhances the glycolytic metabolism (Ray JP et al., Immunity, 2015). The same report found that the mTOR signaling is downstream of CD25. Since CD25 induces mTOR and vice versa (mTOR and pentanoate-mediated glycolysis increases CD25), we suggest a positive feedback between these two molecules. We have included novel data showing that the pharmacological inhibition of glycolysis completely suppresses the pentanoate-induced CD25 expression in CTLs into Supplementary Figure 8. We have also discussed these data and results from the Ray JP et al., Immunity, 2015 paper in the discussion.

Minor suggestions are as follows:

- provide details on human CAR construct, where is tEGFR put into vector.
- add cartoons regarding T cell stimulation (anti-CD3/28 and IL2) in figures 1 and 3.
- why is butyrate not included in Figs 3H/4E; and why have authors (in Figs 1-5) not consistently shown IFNg+, TNFa+ CD8 T cells (rather than sometimes one, sometimes the other, sometimes both cytokines).

- discuss metabolic and/or epigenetic requirements for pentanoate sensitivity; for instance, are T cells that express immune checkpoints potentially less sensitive for pentanoate.
- discuss why pentanoate-stimulation of T cells is preferred over in vivo administration of pentanoate; also discuss the relevance of providing *M. massiliensis* as a probioticum for patients receiving immune therapy.

Author response: We thank the reviewer for these comments. We have now reworked the manuscript to improve clarity and readability.

We have now included a cartoon illustrating the structure of the human CAR construct (Ext. Data Fig. 11).

We thank the reviewer for this nice idea. We have incorporated new cartoons regarding T cell stimulation in Fig.1 and Fig. 3.

We have not included butyrate in all figures, because we mainly focused on the SCFA pentanoate, which is relatively unexplored and unknown bacterial metabolite. It might even be even of an advantage to use pentanoate instead of butyrate, because, in contrast to butyrate, pentanoate is not able to induce Tregs, thus being potentially able to avoid adverse effects.

We have tried to consistently display IFN γ ⁺TNF α ⁺CD8⁺ T cells throughout the manuscript.

We have tried to combine the immune checkpoint blockade together with pentanoate treatment. Unfortunately, in vivo administration of pentanoate did not provide the beneficial effects, with or without the immune checkpoint blockade. We have discussed this important issue in the manuscript. We appreciate all suggestion of the reviewer, and hope that our attempts to answer reviewer's points are satisfactory. We would like to point out that this study (at least to our knowledge) is the first one describing beneficial effects of specific microbial metabolites on the adoptive and CAR T cell therapy, which was the main focus of this study.

REVIEWERS' COMMENTS

Reviewer #4 (Remarks to the Author):

Authors have well-addressed my suggestions, and new results and discussions have improved the quality of the study.